 **eLIFE**

# A combination of transcription factors mediates inducible interchromosomal contacts

**Seungsoo Kim[1], Maitreya J Dunham[1], Jay Shendure[1,2,3]***

[1]Department of Genome Sciences, University of Washington, Seattle, United States; [2]Howard Hughes Medical Institute, Seattle, United States; [3]Brotman Baty Institute for Precision Medicine, Seattle, United States

**Abstract** The genome forms specific three-dimensional contacts in response to cellular or environmental conditions. However, it remains largely unknown which proteins specify and mediate such contacts. Here we describe an assay, MAP-C (Mutation Analysis in Pools by Chromosome conformation capture), that simultaneously characterizes the effects of hundreds of *cis* or *trans*-acting mutations on a chromosomal contact. Using MAP-C, we show that inducible interchromosomal pairing between *HAS1pr-TDA1pr* alleles in saturated cultures of *Saccharomyces* yeast is mediated by three transcription factors, Leu3, Sdd4 (Ypr022c), and Rgt1. The coincident, combined binding of all three factors is strongest at the *HAS1pr-TDA1pr* locus and is also specific to saturated conditions. We applied MAP-C to further explore the biochemical mechanism of these contacts, and find they require the structured regulatory domain of Rgt1, but no known interaction partners of Rgt1. Altogether, our results demonstrate MAP-C as a powerful method for dissecting the mechanistic basis of chromosome conformation.

DOI: https://doi.org/10.7554/eLife.42499.001

## Introduction

The three-dimensional organization of the genome within the nucleus is structured but dynamic (*Bonev and Cavalli, 2016*). Although many features of this conformation are largely conserved across cell types and conditions (*Rao et al., 2014*; *Schmitt et al., 2016a*), some chromatin loops and contacts form specifically in response to signals such as differentiation (*Bonev et al., 2017*; *Monahan et al., 2019*; *Schmitt et al., 2016a*; *Stadhouders et al., 2018*), changes in nutrient availability (*Brickner et al., 2015*; *Brickner et al., 2016*), heat shock (*Chowdhary et al., 2019*; *Chowdhary et al., 2017*), drugs (*D'Ippolito et al., 2018*), meiosis (*Muller et al., 2018*), or circadian rhythms (*Kim et al., 2018*). This dynamic three-dimensional organization of the genome plays a role in regulating gene expression in diverse organisms. In multicellular organisms, active developmental gene promoters form long-range loops with specific enhancer elements (*Bonev et al., 2017*), and this looping is in some cases sufficient for transcriptional activation (*Deng et al., 2014*). In the budding yeast *Saccharomyces cerevisiae*, a well-studied model of genome conformation, genes targeted to nuclear pores are activated (*Taddei et al., 2006*), while those at the nuclear periphery are repressed (*Andrulis et al., 1998*).

Transcription factors (TFs) are attractive candidates for orchestrating such dynamic changes in chromatin conformation, given their site-specific DNA binding and changes in abundance or activity in response to differentiation and cellular signals (*Lambert et al., 2018*). For many conditions, it remains unknown exactly which TFs bind to any given locus. Although binding site motifs are known for many TFs, motif searches poorly predict TF binding (*Guertin and Lis, 2010*; *Jolma et al., 2015*; *Le et al., 2018*; *Levo et al., 2015*; *Liu et al., 2006*; *Slattery et al., 2014*). Even if the set of TFs

*For correspondence:
shendure@uw.edu

Competing interests: The authors declare that no competing interests exist.

**eLife digest** Inside cells, genetic information is stored within molecules of DNA that are folded into three-dimensional structures known as chromosomes. Each fold in a chromosome forms when two points on a single DNA molecule link together to make a loop. DNA in two different chromosomes can also form links with each other (known as "contacts"). Many cells contain two copies of every chromosome and these copies are often able to make contacts with each other.

DNA loops and contacts can change in response to the environment and this may help cells switch the right genes on and off at specific times. For example, in budding yeast cells that have used up most of their preferred food source – a sugar called glucose – the two copies of a region of DNA known as the *HAS1pr-TDA1pr* region stick together. This may help the budding yeast cells switch on genes that are needed to make use of alternative sources of food.

Cells contain hundreds of proteins called transcription factors that can bind to specific locations on DNA and can also stick to each other. These proteins are thought to be responsible for anchoring bridges between the DNA at most loops and contacts. One way to find out which transcription factors form specific DNA loops and contacts is to generate many different genetic mutations in the DNA and identify precisely which mutations disrupt the links. However, current methods can only test one mutation at a time, so it remains unclear how and why many segments of DNA stick together.

Now, Kim et al. have developed a new method known as MAP-C to test how hundreds of mutations in budding yeast affect a particular DNA contact, in a single experiment. The MAP-C method was used to test which mutations within either the DNA segment involved in the contact, or in genes encoding transcription factors, prevent copies of the *HAS1pr-TDA1pr* region from forming contacts. This revealed that three transcription factors – Leu3, Sdd4, and Rgt1 – bridge contacts between the two copies of *HAS1pr-TDA1pr*.

Mutations that disrupt the three-dimensional structure of chromosomes can cause cancer, developmental disorders and other diseases. The MAP-C method will allow researchers to better understand which transcription factors control how DNA is folded inside the cell, and which mutations change this folding.

DOI: https://doi.org/10.7554/eLife.42499.002

bound to each locus is known, it is unclear which TFs are capable of forming chromosomal contacts. DNA-bound TFs can also recruit other cofactor proteins that can mediate chromosomal contacts (*Deng et al., 2012*; *Monahan et al., 2019*; *Song et al., 2007*), but our understanding of TF-cofactor interactions remains incomplete.

Among chromosomal contacts and loops, interchromosomal contacts are less well-understood. This is in part due to the relative paucity of interchromosomal contacts in Hi-C and other 3C (chromosome conformation capture) data, which results from their greater contact distance (*Maass et al., 2018*) and chromosomal self-association into territories (*Cremer and Cremer, 2010*). Nevertheless, many distinct classes of interchromosomal contacts are known, including clustering of transcriptionally active genes (*Mitchell and Fraser, 2008*; *Osborne et al., 2004*; *Schoenfelder et al., 2010*), associations with nuclear bodies (*Quinodoz et al., 2018*), interactions among developmental enhancers and promoters (*Lomvardas et al., 2006*; *Monahan et al., 2019*), and mitotic homologous chromosome pairing in organisms ranging from yeast (*Burgess et al., 1999*) to flies (*Henikoff and Dreesen, 1989*; *Joyce et al., 2016*; *Morris et al., 1999*) and mammals (*Xu et al., 2006*). However, our understanding of the molecular mechanisms of these contacts remains incomplete.

Many known mechanisms for establishing 3D chromosome conformation may act on both intrachromosomal loops and interchromosomal contacts. The emerging consensus model for such DNA-DNA interactions involves loop extrusion by cohesin and other Structural Maintenance of Chromosomes (SMC) factors, which is thought to primarily mediate intrachromosomal loops (*Alipour and Marko, 2012*; *Rao et al., 2014*; *Rowley and Corces, 2018*; *Sanborn et al., 2015*; *Swygert et al., 2019*). However, SMC complexes are also capable of mediating interactions between multiple DNA molecules, such as between sister chromatids (*Michaelis et al., 1997*). Interchromosomal contacts (*Monahan et al., 2019*) and intrachromosomal contacts (*Weintraub et al., 2017*; *Deng et al., 2012*)

can also be mediated by dimerization of structured proteins. In addition to these well-defined strong molecular interactions, weak interactions such as those underlying phase separation of nuclear factors such as transcription factors (*Boija et al., 2018*; *Chong et al., 2018*), coactivators (*Cho et al., 2018*), RNA polymerase (*Boehning et al., 2018*), and heterochromatin proteins (*Larson et al., 2017*; *Strom et al., 2017*) may play an important role in shaping 3D genome organization. How these and other mechanisms synergize remains an open question.

Mitotic (or somatic) homologous chromosome pairing is the preferential association of homologous pairs of loci in mitotically dividing cells. Homolog pairing occurs along the length of the genome in *Drosophila* (*Joyce et al., 2016*), but is more subtle in yeast and other organisms, where the association is often transient and/or genomically localized (*Xu et al., 2006*). Fluorescence in situ hybridization screens in flies have nominated various pairing and anti-pairing factors that modulate the strength of homolog pairing (*Joyce et al., 2012*), but the precise mechanisms by which these factors regulate pairing are largely unknown. In mammals, X chromosome pairing is mediated by CTCF and Oct4 (*Donohoe et al., 2009*), in conjunction with transcription (*Xu et al., 2007*). However, cases of highly localized homolog pairing remain rare. Furthermore, the distinctions between homolog pairing and non-allelic interactions between repetitive elements (*Gladyshev and Kleckner, 2017*; *Mirkin et al., 2014*) remain unclear.

We recently identified a novel example of an inducible, localized interchromosomal contact between homologous copies of the *HAS1pr-TDA1pr* locus in diploid *Saccharomyces* yeasts (*Kim et al., 2017*). This interaction occurs in saturated culture conditions, requires the 1 kb intergenic region between the *HAS1* and *TDA1* coding sequences, and is detectable by both Hi-C and microscopy. The condition-specificity and dependence on intergenic sequence led us to hypothesize that one or more TFs might mediate this pairing. Although yeast TF binding is well-characterized for standard growth conditions (*Badis et al., 2008*), TF binding has not been systematically measured in saturated culture conditions. Meanwhile, computational predictions of TF binding sites are insufficiently specific: for example, within the 1 kb region required for *HAS1pr-TDA1pr* pairing, dozens of TFs have at least one motif match. Furthermore, even if the TFs bound to this region were known, it would remain unclear which subset played a role in mediating inducible interchromosomal pairing.

Here we describe a method that enables the simultaneous testing of hundreds of *cis* or *trans*-acting mutations for their effects on a chromosomal contact of interest. As a proof of concept, we applied this method, which we call Mutation Analysis in Pools by Chromosome conformation capture (MAP-C), to characterize the molecular components mediating *HAS1pr-TDA1pr* pairing. We first perform saturating mutagenesis of the regulatory region that mediates the interchromosomal pairing (*cis* MAP-C) to identify sequence motifs required for pairing that potentially correspond to TF binding sites. We then test the effects of knocking out over one hundred TFs (*trans* MAP-C), and confirm that three—Leu3, Sdd4 (Ypr022c), and Rgt1—are necessary for inducible interchromosomal pairing. We verify their binding by chromatin immunoprecipitation, and find that *HAS1pr-TDA1pr* exhibits the strongest coincident, combined binding by all three factors across the genome in a condition-specific manner. We further use *trans* MAP-C to interrogate how interaction partners and domains of Rgt1 regulate pairing. Finally, we make an initial attempt to characterize the functional consequences of *HAS1pr-TDA1pr* pairing. Taken together, our results demonstrate how a combination of TFs can mediate inducible interchromosomal pairing. Furthermore, our study shows the utility of a pooled mutant approach to studying both the *cis* and *trans* dependencies of chromosome conformation.

## Results

### A pooled approach to systematically dissect chromosome conformation

In order to identify and dissect the molecular mechanisms underlying chromosome conformation, experiments involving perturbations (*e.g.* mutations) are needed. However, despite numerous advances in chromosome conformation capture (3C) technology over the last two decades (*de Wit and de Laat, 2012*; *Schmitt et al., 2016b*), each experiment characterizes a single sample, which limits the number of genes or *cis*-regulatory elements that can be disrupted (*Monahan et al., 2019*; *Nora et al., 2017*; *Schwarzer et al., 2017*; *Weintraub et al., 2017*).

To address this limitation and enable systematic screens, we developed MAP-C, an assay in which hundreds of mutations are simultaneously tested for their effects on a single chromosomal contact of interest (*Figure 1A*). In the *cis* version of MAP-C, which we describe first, these mutations are targeted to one of the regions involved in the chromosomal contact. In the *trans* version of MAP-C, mutations can be spread across the genome, as long as they are associated with a unique barcode sequence at the chromosomal contact site.

The first step of *cis* MAP-C is to generate an allelic series of a region of interest, which can be achieved in a cost-effective manner via array-synthesized oligonucleotide pools or error-prone PCR. Alternatively, if desired, variants can be generated individually and then pooled prior to conducting MAP-C. The resulting mutant pool is then integrated into the genome and subjected to the 3C assay (*Dekker et al., 2002*). The region containing the genetic variants is amplified using two different primer pairs: the first (3C library) amplifies a specific ligation product, and the second (genomic library) amplifies regardless of ligation. These amplification products are deeply sequenced to measure the abundance of each variant in the 3C library, which is normalized to its abundance in the genomic library. The relative extent to which sequence variants participate in the chromosomal contact of interest is proportional to their normalized representation in the 3C library.

## A cluster of TF motifs is necessary and sufficient for *HAS1pr-TDA1pr* pairing

As a first test of *cis* MAP-C, we sought to systematically dissect the conserved pairing between *HAS1pr-TDA1pr* homologs in diploid *Saccharomyces* yeasts grown to saturation. We recently used Hi-C of *S. cerevisiae* x *S. uvarum* hybrids (<80% nucleotide identity) to discover this homolog pairing interaction, and furthermore identified a 1,038 bp noncoding region that was necessary and sufficient for pairing (*Kim et al., 2017*).

To find a minimal subsequence of the 1,038 bp *HAS1pr-TDA1pr* region that is sufficient to pair with other *HAS1pr-TDA1pr* alleles, we replaced the native *S. cerevisiae HAS1pr-TDA1pr* locus with a library containing each of 861 tiling 178 bp subsequences of the 1,038 bp region (along with a G418 resistance cassette and restriction site), in *S. cerevisiae* x *S. uvarum* hybrid yeast. We then performed *cis* MAP-C for pairing of the modified locus with the *S. uvarum* copy of *HAS1pr-TDA1pr* on a saturated culture of the pool, in two replicates (*Figure 1B*). Compared to the genomic libraries, the 3C libraries were highly enriched for a narrow region spanning ~ 500 to ~ 700 bp from the *HAS1* coding sequence, with a plateau between ~ 525 to ~ 675 bp, consistent with that ~ 150 bp region being the only subsequence shorter than 178 bp sufficient for pairing (termed the 'minimal pairing region' below). To confirm that this pattern of enrichment is specific to *HAS1pr-TDA1pr* homolog pairing rather than underlying all of its chromosomal contacts, we repeated the assay with a pair of primers amplifying a ~ 10 kb intrachromosomal contact (*Figure 1—figure supplement 1A and B*). In this 'off-target' control, coverage from the 3C library matched that of the genomic control, suggesting that most variants are capable of intrachromosomal looping, but only those containing the minimal pairing region are capable of interchromosomal pairing with the other *HAS1pr-TDA1pr* allele.

Since our initial experiment was performed in the endogenous genomic context, other DNA sequences outside but near the *HAS1pr-TDA1pr* locus could be required in addition to the minimal pairing region. We therefore tested whether inserting a 184 bp sequence that included the minimal pairing region into an ectopic location, the gene *FIT1* (*YDR534C*), would induce pairing with the native *HAS1pr-TDA1pr* locus in saturated cultures of haploid *S. cerevisiae* (*Figure 1—figure supplement 1C*). As a negative control, we inserted an equivalently sized subsequence insufficient for pairing into the same locus (*Figure 1—figure supplement 1C*). Indeed, insertion of the minimal pairing region led to a > 30 fold increase in 3C signal for pairing with *HAS1pr-TDA1pr* as compared to the negative control (*Figure 1C*).

We next sought to obtain a base-pair resolution map of the DNA sequences necessary for pairing. We used error-prone PCR to generate variants of a 207 bp region (161 bp excluding fixed primer sequences) containing the minimal pairing region, with an average of 1.49 substitutions (range 0–14) per template (*Figure 1—figure supplement 1D*). We inserted this variant library in place of the native *S. cerevisiae HAS1pr-TDA1pr* sequence as before, and performed *cis* MAP-C. The ratio of total substitution abundance in the 3C and genomic libraries can be plotted at each mutagenized position. This identified six clusters of two or more adjacent positions showing strong depletion of substitutions in the 3C libraries, indicating that they are required for *HAS1pr-TDA1pr*

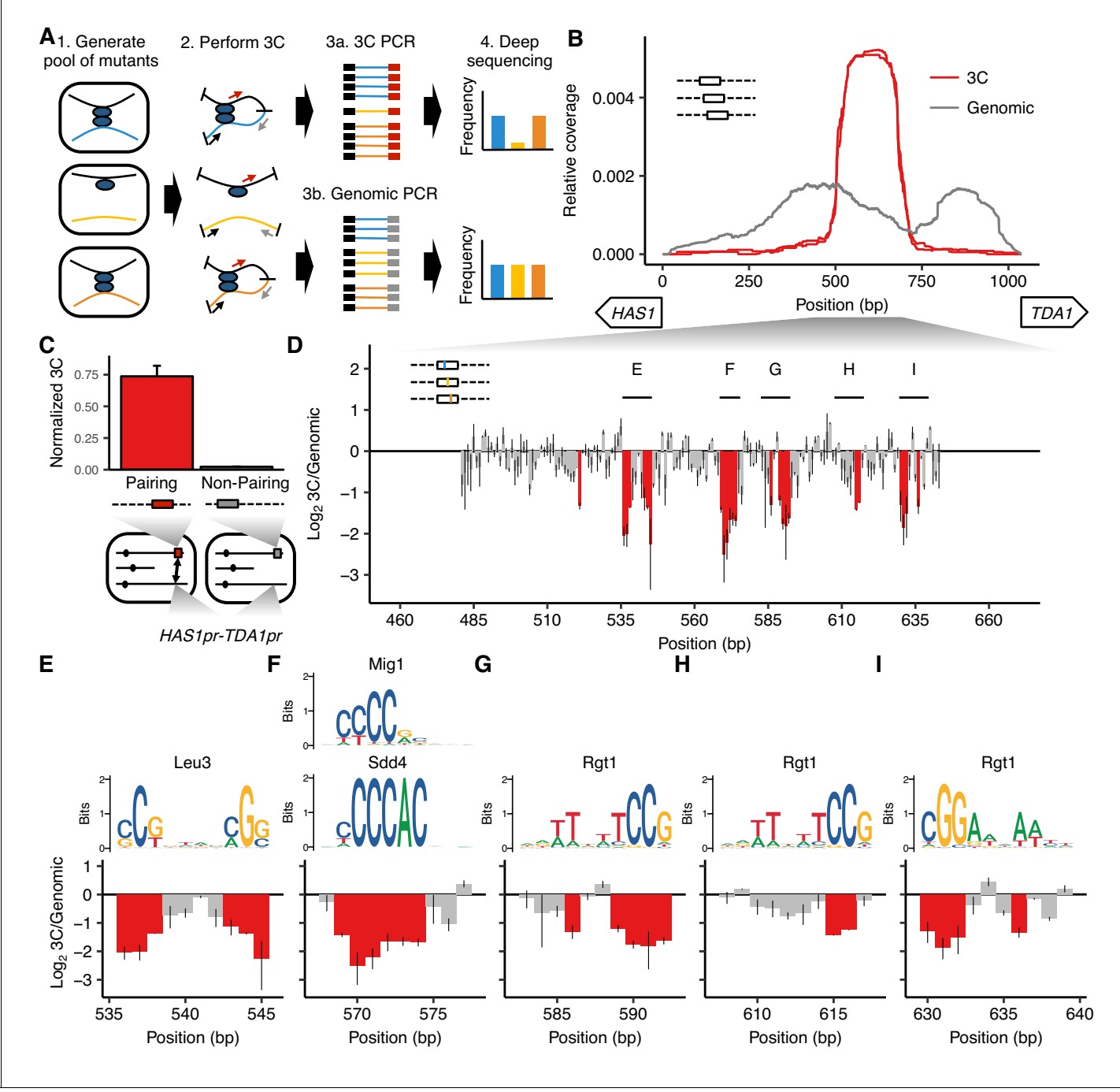

**Figure 1.** MAP-C identifies DNA sequences necessary and sufficient for inducible pairing between *HAS1pr-TDA1pr* alleles. (**A**) In the *cis* MAP-C method, mutations in a ~ 250 bp segment of the genome are assessed for their effect on a specific 3D contact of that segment. Colored lines indicate mutant DNA sequences, and thin arrows indicate primers. (**B**) A ~ 150 bp region is sufficient for interchromosomal pairing. *cis* MAP-C was used to test 178 bp subsequences from the *S. cerevisiae HAS1pr-TDA1pr* region for pairing with the *S. uvarum HAS1pr-TDA1pr*. Shown are read coverage of the 3C (red) and genomic (gray) libraries, normalized to sum to 1. The two lines for each color represent technical replicates. Start positions and orientations of *HAS1* and *TDA1* coding sequences are shown on x-axis. (**C**) A minimal pairing region is sufficient for ectopic pairing. Shown are contact frequencies between *HAS1pr-TDA1pr* and a pairing (red) or non-pairing (gray) sequence (coordinates shown in *Figure 1—figure supplement 1C*) integrated at the *FIT1* locus in haploid *S. cerevisiae*, as measured by 3C, normalized to contacts between *FIT1* and *HLR1* 10 kb away. Bars indicate mean ± s.d. of technical triplicates. (**D**) Base-pairs necessary for pairing, shown as ratio of the total substitution frequency at each position in the 3C library compared to the genomic library. Error bars indicate the two technical replicates. Positions most strongly required for pairing (log₂3C/Genomic < −1.1) are

*Figure 1 continued on next page*

*Figure 1 continued*

highlighted in red. (**E–I**) Selected regions from panel D are highlighted, with sequence logos for matching transcription factor motifs. See ***Figure 1—figure supplements 2*** and ***3*** for full set of overlapping motifs.

DOI: https://doi.org/10.7554/eLife.42499.003

The following figure supplements are available for figure 1:

**Figure supplement 1.** Design and controls for using *cis* MAP-C to dissect *HAS1pr-TDA1pr* pairing.

DOI: https://doi.org/10.7554/eLife.42499.004

**Figure supplement 2.** Motifs overlapping positions required for *HAS1pr-TDA1pr* pairing.

DOI: https://doi.org/10.7554/eLife.42499.005

**Figure supplement 3.** Lower-scoring motifs matching positions required for *HAS1pr-TDA1pr* pairing.

DOI: https://doi.org/10.7554/eLife.42499.006

**Figure supplement 4.** Validation of TF motifs required for *HAS1pr-TDA1pr* pairing with a 3 bp substitution mutant library.

DOI: https://doi.org/10.7554/eLife.42499.007

**Figure supplement 5.** Conservation of TF motifs in *HAS1pr-TDA1pr*.

DOI: https://doi.org/10.7554/eLife.42499.008

pairing (***Figure 1D***). We inspected motifs in the region to identify candidate TFs that might mediate the pairing (***Figure 1—figure supplements 2*** and ***3***). Given the abundance of potential matches, we prioritized motifs with high-scoring matches, motifs with conserved positions corresponding to those most important for pairing, and motifs occurring in multiple clusters. The first two clusters together aligned to a Leu3 motif (***Figure 1E***), the third to several similar motifs, including Sdd4 (Ypr022c) and Mig1 (***Figure 1F***), and the last three to Rgt1 motifs in both orientations (***Figure 1G–I***). These motifs, which span 47 bp, include 23 of the 24 positions most depleted for mutations in the 3C libraries (***Figure 1—figure supplement 1E***), and all clusters of two or more adjacent such positions (***Figure 1D***). None of these mutations had a strong effect on intrachromosomal looping (***Figure 1—figure supplement 1F***), and all of the clusters were reproduced using an alternative mutagenesis strategy (programmed 3 bp substitutions) (***Figure 1—figure supplement 4***). Interestingly, a fourth Rgt1 motif and a second Sdd4/Mig1 motif mutagenized only in our validation experiment were not required for pairing, suggesting that either not all motifs in this region are bound by the same TFs or not all bound TFs are involved in mediating homolog pairing at this locus.

Thus, using *cis* MAP-C, we identified a ~ 150 bp subsequence of *HAS1pr-TDA1pr* sufficient for pairing, containing five required TF motif occurrences. If these TF motifs are together sufficient for pairing, we would expect that 1) they are only observed in a cluster in the minimal pairing region and not elsewhere in the *HAS1pr-TDA1pr*, and 2) they are present in a cluster in the *S. uvarum* copy of this region, and potentially other *Saccharomyces* as well. Indeed, these motifs are clustered together only in the central region of *HAS1pr-TDA1pr*; remarkably, this pattern holds across all *Saccharomyces* species (***Figure 1—figure supplement 5***).

## Three transcription factors are required for pairing

Although we identified the TF motifs required for *HAS1pr-TDA1pr* homolog pairing at base-pair resolution with *cis* MAP-C, the redundancy among TF motifs made it difficult to definitively identify the TFs involved (***Figure 1F***). To address this, we developed a modified version of MAP-C that is capable of assaying *trans* mutations spread across the genome, such as gene knockouts, for their effects on a specific chromosomal contact (*trans* MAP-C). With *trans* MAP-C, each mutation is uniquely associated with a short barcode sequence near a region involved in the chromosomal contact of interest, which is then assayed by 3C. Mutations that affect pairing frequency modulate the abundance of their corresponding barcodes in the 3C products, which can be readily quantified by deep sequencing. To study the *trans* requirements of *HAS1pr-TDA1pr* pairing, we ectopically integrated barcoded versions of the minimal pairing region, wherein the barcode identifies which TF is knocked out, and assayed their capacity to pair with the native copy of *HAS1pr-TDA1pr* (***Figure 2A***).

We first tested this approach in a pilot set of 10 TF knockouts by inserting the minimal pairing region (***Figure 1—figure supplement 1C***) into the common KanMX drug resistance cassette used to replace each deleted gene in the haploid yeast deletion collection (***Giaever et al., 2002***). We then assayed these constructs for interactions with the native *HAS1pr-TDA1pr* region, using the existing gene-specific barcodes to measure strain abundances in each library. In this approach, because the

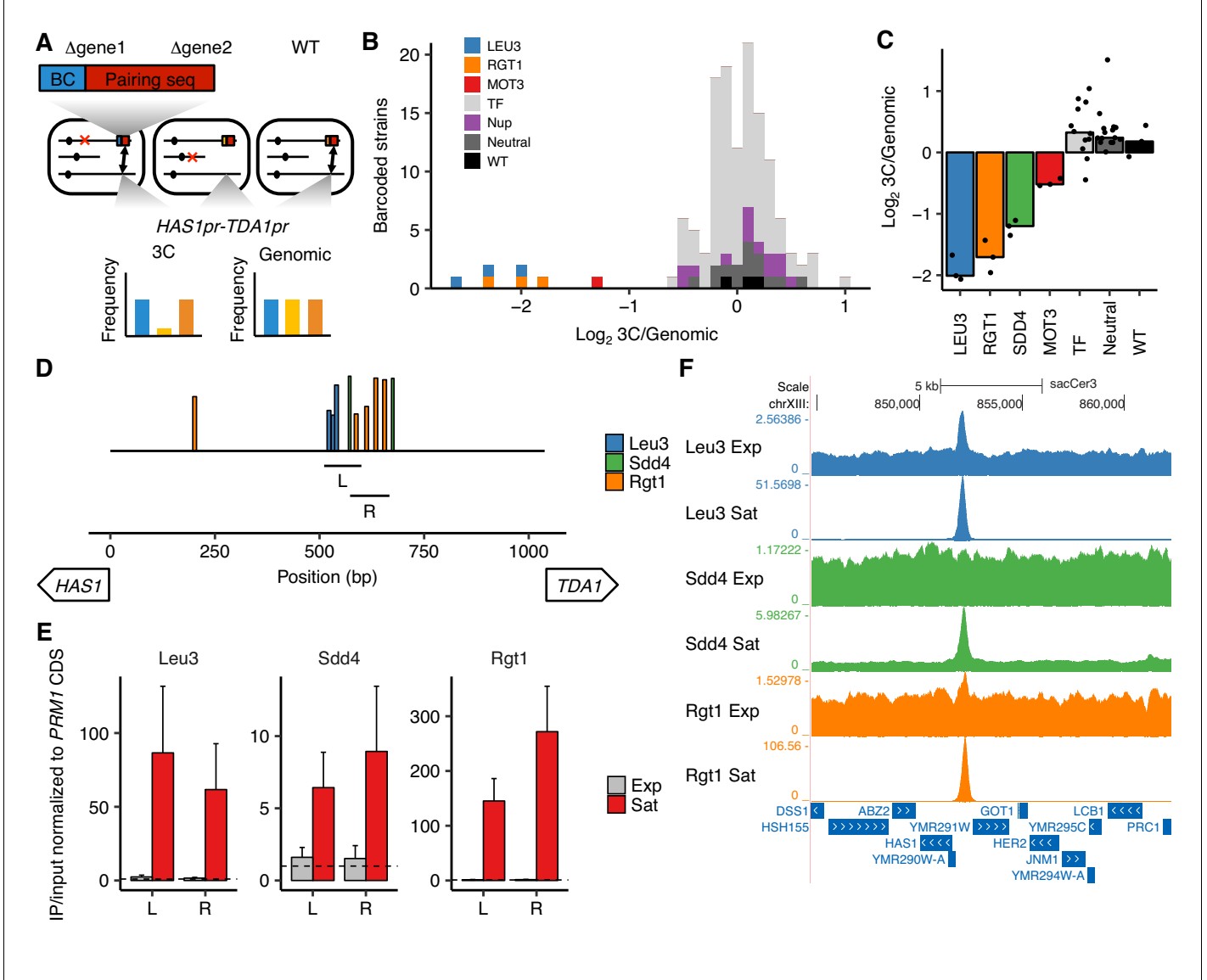

**Figure 2.** Transcription factors Leu3, Sdd4, and Rgt1 mediate *HAS1pr-TDA1pr* pairing. (A) In trans MAP-C, the effect of TF knockouts or variants are assessed for their effect on a specific 3D contact by association with barcodes. Barcoded versions of the minimal pairing sequence were ectopically integrated and assessed for pairing with the native *HAS1pr-TDA1pr* in haploid *S. cerevisiae*. Red Xs indicate gene knockouts; red boxes indicate pairing sequence (*Figure 1—figure supplement 1C*); BC indicates barcode; double-headed arrows indicate the presence of chromosomal contacts. (B) Full TF knockout screen identifies Leu3 and Rgt1 as *trans* requirements for pairing. Histogram of relative abundance of each barcoded gene knockout strain in 3C library compared to the genomic library, excluding strains below a frequency of 0.3% of the pool. Barcode replicates are shown as separate squares in histogram. *LEU3*, *RGT1*, and *MOT3* are highlighted individually; TF indicates other transcription factors; Nup indicates nuclear pore complex components; Neutral indicates fitness-neutral negative controls (*Figure 2—figure supplement 2A*). (C) Validation TF knockout screen confirms that Leu3, Sdd4, and Rgt1 are required for pairing. Bar plot of median relative abundance in 3C library compared to the genomic library, with overlaid scatter plot of individual barcoded strains. TF includes *MIG1*, *VHR1*, *CBF1*, and *YGR067C*. (D) Regions tested by ChIP-qPCR for TF binding. Bars show Leu3, Sdd4, and Rgt1 motif matches in *S. cerevisiae HAS1pr-TDA1pr*, and lines indicate regions (L and R) used for qPCRs in panel E. Bar heights indicate motif score as fraction of maximum possible score; all motifs with a score of at least 0.45 shown. (E) TFs bind *HAS1pr-TDA1pr* more strongly in saturated conditions. Chromatin immunoprecipitation qPCR results in exponentially growing (Exp) and saturated (Sat) cultures, normalized by input and to a negative control locus in the *PRM1* coding sequence. L and R indicate two primer sets as shown in panel D. Bars indicate mean ± s.e.m. of biological triplicates. Dashed lines indicate a value of 1 (background enrichment). (F). Chromatin immunoprecipitation sequencing data near the *HAS1pr-TDA1pr* locus (coordinates chrXIII:844,705–862,314 in sacCer3 reference), shown as fold enrichment in IPs over input.

DOI: https://doi.org/10.7554/eLife.42499.009

The following figure supplements are available for figure 2:

**Figure supplement 1.** A pilot TF gene knockout screen for *HAS1pr-TDA1pr* pairing.

*Figure 2 continued on next page*

*Figure 2 continued*

DOI: https://doi.org/10.7554/eLife.42499.010

**Figure supplement 2.** An expanded *trans* knockout screen for *HAS1pr-TDA1pr* pairing.

DOI: https://doi.org/10.7554/eLife.42499.011

**Figure supplement 3.** ChIP-seq of Leu3, Sdd4, and Rgt1 shows stronger motif-driven binding in saturated conditions.

DOI: https://doi.org/10.7554/eLife.42499.012

**Figure supplement 4.** De novo motif discovery reveals ChIP-seq enrichment of known motifs, poly-T tracts enriched in promoters, and tRNA genes.

DOI: https://doi.org/10.7554/eLife.42499.013

pairing sequences are inserted into different genes throughout the genome, the frequency of pairing is confounded by the potential effects of the genomic location of the pairing sequence. Therefore, for each of the 10 TFs we targeted, we included as controls up to six neighboring genes, which should have a similar genomic location effect as the targeted gene (*Figure 2—figure supplement 1A and B*). We hypothesized that due to the Rabl orientation of yeast chromosomes, in which centromeres are clustered together (*Duan et al., 2010*), centromere-proximal regions would interact less with *HAS1pr-TDA1pr*, which is centromere-distal. Indeed, the most centromere-distal gene knockouts interacted ~ 4 fold more with the *HAS1pr-TDA1pr* locus than the most centromere-proximal gene knockouts (*Figure 2—figure supplement 1C*). Of the 7 TF gene knockouts that were measured in our assay (three dropped out during library construction, including *LEU3*), six had no substantial difference in pairing strength compared to their genomic neighbors (*Figure 2—figure supplement 1D*). However, deletion of *RGT1* led to a ~ 20 fold decrease in pairing strength, consistent with Rgt1 binding its cognate motifs in the minimal pairing region (*Figure 2B and C*).

Next, we expanded our *trans* MAP-C screen to include the majority of known nonessential TFs with known binding motifs (*de Boer and Hughes, 2012*). To avoid the confounding effect of genomic location, we inserted a barcoded pairing sequence construct into a fixed locus instead of into the gene knockout locations. We associated each of these barcodes with the cognate knockout by individually transforming each knockout strain with a unique barcode, in a 96-well plate format. We tested a total of 109 TF gene knockouts, as well as 15 nuclear pore complex components, eight fitness neutral negative controls, and a wild-type control, with multiple barcode replicates for controls and expected hits (*Figure 2—figure supplement 2A*). As expected, most barcoded strains were equally abundant in the 3C and genomic libraries, indicating that the corresponding knockout did not impact *HAS1pr-TDA1pr* pairing. However, *LEU3* and *RGT1* knockouts were depleted ~ 4 fold from the 3C libraries, suggesting that they are required for pairing (*Figure 2B*). In addition, deletion of *MOT3* modestly decreased pairing (~2.5 fold); however, its binding motif is not present in the *HAS1pr-TDA1pr* region, suggesting an indirect role. Two other knockouts, *VHR1* and *CBF1*, appeared to also decrease pairing, but were at low abundances and might reflect noise (*Figure 2—figure supplement 2B*).

Surprisingly, none of the TFs required for pairing in either *trans* MAP-C experiment had a motif matching the sequence CCCCAC (the third cluster of positions required for pairing; *Figures 1F* and *2B*, and *Figure 2—figure supplement 1C*). However, two putative TFs with high scoring motif matches, *YPR022C* (*SDD4*) and *YGR067C*, were excluded in the initial screens due to their lack of annotations. Therefore, we repeated our fixed-locus TF knockout screen with a limited set of genes, including the two putative TFs and additional replicates for the hits *MOT3*, *VHR1*, and *CBF1*. We found that *SDD4* is indeed required for pairing, suggesting that it is the *trans*-acting factor that binds the CCCCAC motif (*Figure 2C*). *MOT3* once again exhibited modest depletion, suggesting a minor, perhaps indirect, role in pairing, whereas *VHR1* and *CBF1* displayed no depletion (*Figure 2C*).

The strong concordance between the genomic base-pairs required for pairing (identified by *cis* MAP-C) and the DNA binding motifs of *trans* factors required for pairing (identified by *trans* MAP-C) suggests that Leu3, Sdd4, and Rgt1 bind the *HAS1pr-TDA1pr* to mediate pairing. To test this hypothesis, we performed chromatin immunoprecipitation (ChIP) for the tandem affinity purification (TAP) tagged versions of Leu3, Sdd4, and Rgt1 (*Ghaemmaghami et al., 2003*), in haploid *S. cerevisiae* yeast under both saturated and exponential growth conditions. By qPCR using two different primer pairs, all three TFs strongly bound to the *HAS1pr-TDA1pr* pairing region in saturated conditions, but near background levels in exponential growth (*Figure 2D and E*). We then performed

ChIP sequencing to determine TF binding genome-wide. Consistent with our quantitative PCR measurements, all three TFs showed robust ChIP-seq peaks at *HAS1pr-TDA1pr* in saturated conditions. In contrast, only Leu3 demonstrated a significant peak in exponential growth, with 20-fold weaker enrichment (2.5-fold vs. 51-fold in saturated conditions; *Figure 2F*). For all three TFs, saturated conditions produced ChIP-seq peaks with greater enrichments over the input controls and more robust enrichment of the expected motifs, suggesting generally more extensive DNA binding (*Figure 2—figure supplements 3* and *4*). These global trends could be a result of technical artifacts as well as biological differences; however, all samples were treated identically using a protocol not optimized for saturated culture conditions, and the *HAS1pr-TDA1pr* peak showed particularly strong condition-specificity (*Figure 2—figure supplement 3*).

Based on the convergence of *cis* MAP-C, *trans* MAP-C, and ChIP-seq data, we conclude that Leu3, Rgt1, and Sdd4 directly bind to both alleles of the *HAS1pr-TDA1pr* minimal pairing region under saturated growth conditions and thereby mediate inducible interchromosomal contacts between them.

## Combinatorial transcription factor binding specifies strong pairing

We next explored whether the combination of Leu3, Sdd4, and Rgt1 binding explains the uniqueness of *HAS1pr-TDA1pr* pairing. If the clustered binding of Leu3, Sdd4, and Rgt1 is necessary and sufficient to cause pairing in saturated culture conditions, either 1) no loci other than *HAS1pr-TDA1pr* should have all three TFs bound on both the *S. cerevisiae* and *S. uvarum* copies, or 2) other loci that do have all three TFs bound should also exhibit pairing. Because DNA binding data was only available for the *S. cerevisiae* genome, we tested the first possibility by scanning the *S. cerevisiae* and *S. uvarum* genomes for clusters of the three motifs using permissive thresholds for motif matches and allowing up to 200 bp between motifs (see Materials and methods), and then assessed ChIP-seq data at these clusters. The promoters of four genes, *TDA1* but also *HXT3*, *YKR075C*, and *SKS1*, harbored a motif cluster containing all three motifs in both *S. cerevisiae* and *S. uvarum* (*Figure 3A*). We also assessed two additional loci, *MIG1pr* and *ILV2pr*, that lacked one of the three motifs in *S. uvarum*. Of these six loci, *TDA1pr* exhibited the strongest total ChIP-seq signal in saturated conditions (*Figure 3B and C*), even when extending the search to motif clusters with only one or two of the three motifs in *S. cerevisiae* (*Figure 3—figure supplement 1*). Furthermore, *YKR075Cpr* was the only other locus with robust binding of all three TFs in *S. cerevisiae*, but the only Leu3 motif in the *S. uvarum* copy of the region overlaps a stronger Rgt1 motif and may not result in Leu3 binding. We also noticed that the *TDA1pr* cluster was the most compact (i.e. shortest maximum distance between motifs); whether this plays a role in pairing is unclear. We infer that the combinatorial binding of Leu3, Sdd4, and Rgt1 on both homologs specifies inducible homolog pairing in saturated cultures.

Although *TDA1pr* exhibits the strongest combinatorial TF binding, we wondered whether any other motif clusters pair inducibly like the *HAS1pr-TDA1pr* locus, albeit perhaps more weakly. To address this question, we leveraged our previously published Hi-C datasets (*Kim et al., 2017*), along with new Hi-C experiments for the high-pairing strain background in which we performed our pooled mutant experiments, to compare the strength of pairing at each homologous motif cluster and assess whether this pairing is condition-specific (*Figure 3—figure supplement 2*). We observed that *HXT3pr* appears to form inducible homologous contacts in saturated culture conditions (*Figure 3D*), despite weak Sdd4 and no Leu3 binding. We did not detect pairing at any other loci (*Figure 3—figure supplement 2*). Across several strain backgrounds, the *HXT3* promoters exhibited 1.6- to 2.7-fold increased interaction frequencies compared to other similar interchromosomal pairs of loci (at least 480 kb from a centromere and excluding subtelomeric regions) in saturated culture conditions (*Figure 3E*), but only at baseline levels during exponential growth in rich medium. This is weaker than the pairing between *HAS1pr-TDA1pr* alleles (*Figure 3—figure supplement 2*), suggesting that combinatorial TF binding is not strictly necessary for inducible pairing but may facilitate particularly strong pairing.

The identification of two pairs of loci, *HAS1pr-TDA1pr* and *HXT3pr*, exhibiting homolog pairing opened the possibility that there might be cross-pairing (nonhomologous contacts) between *HAS1pr-TDA1pr* and *HXT3pr*. To test this possibility, we extracted the interaction frequencies among the four loci (*i.e.* two pairs) from our Hi-C data. In all saturated culture datasets where homolog pairing was present, inter-locus interactions were substantially weaker and similar in frequency to

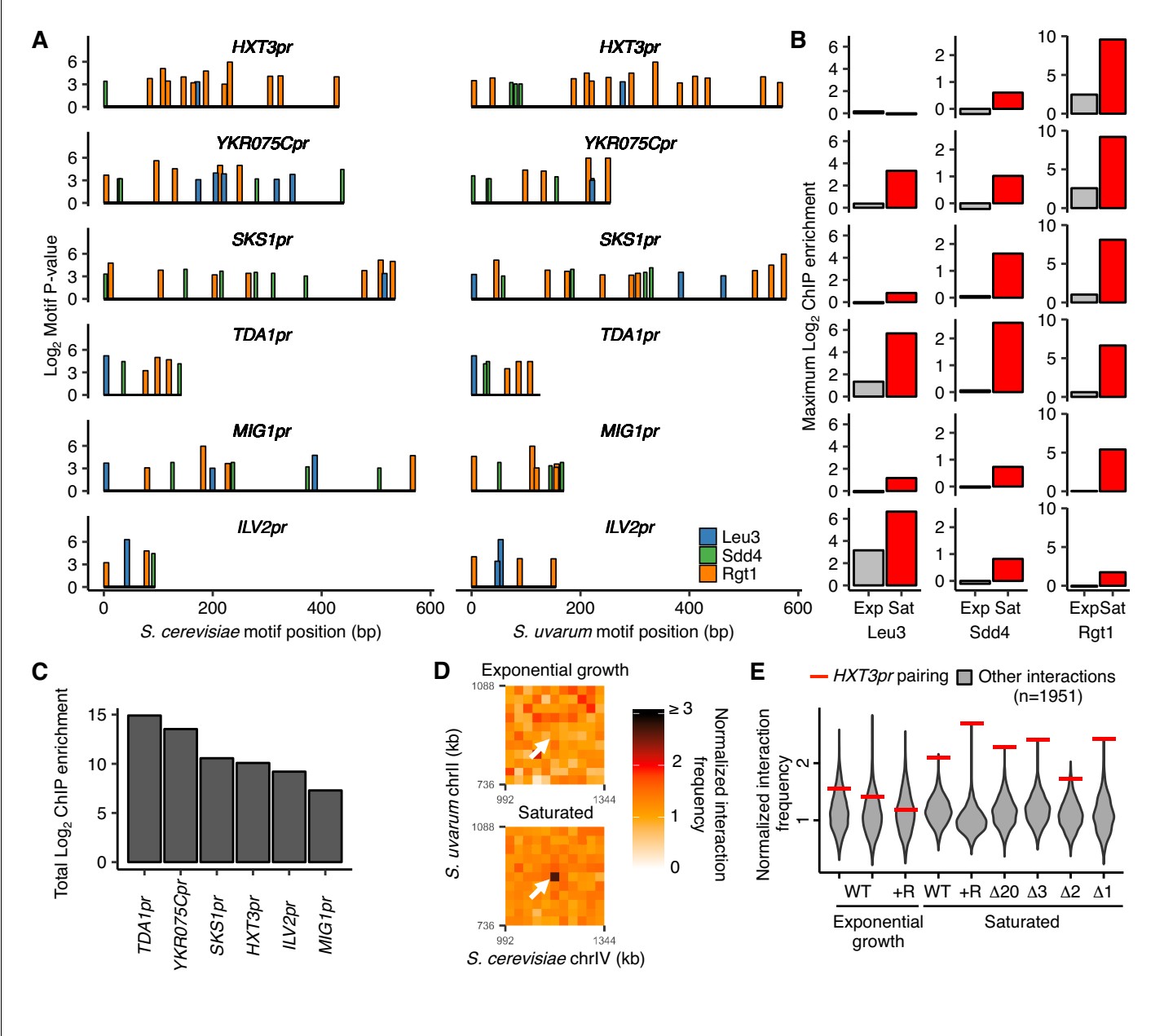

**Figure 3.** Combinatorial TF binding specifies inducible homolog pairing. (**A**) Top clusters of Leu3, Sdd4, and Rgt1 motifs in homologous loci of both *S. cerevisiae* and *S. uvarum* genomes containing all three motifs in *S. cerevisiae*, in order of lowest *P*-value from top to bottom. (**B**) Not all motif clusters are bound by all three TFs. ChIP-seq data for exponentially growing (Exp) and saturated (Sat) *S. cerevisiae* corresponding to motif clusters in panel A. (**C**) Combined TF binding is strongest at *TDA1pr*. Shown are the sum of the ChIP $\log_2$ fold enrichments over inputs in saturated conditions for motif clusters in panel A. (**D**) *HXT3pr* also exhibits inducible homolog pairing. Hi-C contact maps of interactions between regions centered on *S. cerevisiae HXT3* and *S. uvarum HXT3* at 32 kb resolution in exponentially growing and saturated cultures of a *S. cerevisiae* x *S. uvarum* hybrid strain (YMD3920). White arrows indicate interactions between homologous *HXT3* promoters. (**E**) Strength of *HXT3* promoter pairing across conditions and strain backgrounds, at 32 kb resolution (red lines) compared to similar interactions (gray violin plots; i.e. interactions between an *S. cerevisiae* locus and an *S. uvarum* locus, where both loci are $\geq$ 15 genomic bins from a centromere and $\geq$ 2 bins from a telomere, $\geq$ 2 bins from *HAS1pr-TDA1pr*, and not both on chrXII). WT represents strain ILY456 (YMD3259).+R indicates a restriction site added upstream of *HAS1* (YMD3920), Δ20 indicates a 20 kb deletion centered at *S. cerevisiae HAS1* (YMD3266), Δ3 indicates a 3 kb deletion centered at *S. cerevisiae HAS1* (YMD3267), Δ2 indicates a 2 kb deletion of the *S. cerevisiae HAS1* coding sequence (YMD3268), and Δ1 indicates a 1 kb deletion of the *S. cerevisiae HAS1pr-TDA1pr* intergenic region (YMD3269).
DOI: https://doi.org/10.7554/eLife.42499.014

The following figure supplements are available for figure 3:

**Figure supplement 1.** *HAS1pr-TDA1pr* exhibits uniquely strong combinatorial binding of Leu3, Sdd4, and Rgt1.

*Figure 3 continued on next page*

*Figure 3 continued*

DOI: https://doi.org/10.7554/eLife.42499.015

**Figure supplement 2.** Hi-C evidence of inducible homolog pairing at *TDA1* and *HXT3*.

DOI: https://doi.org/10.7554/eLife.42499.016

**Figure supplement 3.** *HAS1pr-TDA1pr* and *HXT3pr* pairing are independent.

DOI: https://doi.org/10.7554/eLife.42499.017

**Figure supplement 4.** Lack of nonhomologous pairing between clusters of Leu3, Sdd4, and Rgt1 motifs.

DOI: https://doi.org/10.7554/eLife.42499.018

those in non-pairing conditions (*Figure 3—figure supplement 1*). Together with our previous experiment demonstrating that two identical copies of the minimal pairing region can pair even at non-allelic locations in haploid *S. cerevisiae* (*Figure 1C*), these data suggest that the interchromosomal pairing mediated by Leu3, Sdd4, and Rgt1 is sequence-specific beyond the simple presence of the same TF binding sites.

## *HAS1pr-TDA1pr* pairing is regulated by Rgt1 abundance, recruitment of Tup1/Ssn6, and competing domains

We next sought to explore the mechanisms that regulate pairing. We hypothesized that TF expression levels might regulate the strength of *HAS1pr-TDA1pr* pairing. To test this hypothesis, we analyzed RNA-seq data for haploid *S. cerevisiae* in pairing and non-pairing conditions (saturated and exponentially growing cultures, respectively) (*Kim et al., 2017*). *RGT1* and *SDD4* were upregulated in saturated cultures, ~ 2 fold and ~ 6 fold, respectively, whereas *LEU3* transcript levels remained constant (*Figure 4A*). These results are consistent with the hypothesis that increased transcription of the TFs mediating pairing, particularly *RGT1*, regulates the strength of the pairing interaction.

Based on these results, we wondered whether overexpression of any one of the three proteins would be sufficient to produce pairing in non-saturated culture conditions. We used the Z$_3$EV estradiol induction system (*McIsaac et al., 2014*) to individually overexpress *S. cerevisiae* Leu3, Sdd4, or Rgt1, and measured pairing between the native *HAS1pr-TDA1pr* loci using 3C in *S. cerevisiae* x *S. uvarum* hybrids growing exponentially in rich medium (*Figure 4B*). In all three strains, a 2 hr estradiol induction led to no increase in pairing strength despite between 2- and 10-fold increases in transcript levels (*Figure 4—figure supplement 1A*). As an alternative test, we used galactose induction to overexpress epitope-tagged *RGT1*, and observed a decrease in pairing strength relative to a strain lacking the overexpression cassette (*Figure 4—figure supplement 1B*). These results are consistent with no single TF being sufficient for pairing; however, Leu3 and Rgt1 are both known to change in conformation (*Sze et al., 1992*) or phosphorylation state (*Kim et al., 2003*) in different conditions, so it remains possible that overexpression of a single TF in the appropriate state suffices for pairing.

Rgt1 is known to interact with several cofactors that affect its DNA binding and transcriptional repression activities: the Tup1/Ssn6 co-repressor complex and the proteins Mth1 and Std1 (*Polish et al., 2005*). Our experiments thus far had not distinguished whether Rgt1's pairing activity is directly mediated by physical interactions among molecules of Rgt1 or indirectly, through these or other interaction partners. To address this, we performed *trans* MAP-C for individual deletions of these four interacting partners of Rgt1, along with the same positive and negative controls as before (*Figure 4C*). In addition, based on the glutamine and asparagine-rich domains present in Sdd4 and Rgt1, we tested deletion of *RNQ1*, a Q/N-rich peptide known to influence the oligomerization of other Q/N-rich proteins (*Derkatch et al., 2004*). Deletion of *RNQ1* had no effect on pairing, suggesting that *HAS1pr-TDA1pr* pairing is not mediated by Q/N-rich domains. Deletion of *TUP1* or *SSN6* both led to increased pairing, indicating that the recruitment of the Tup1/Ssn6 co-repressor complex inhibits pairing. This is consistent with its known inhibition of Rgt1's DNA binding activity (*Roy et al., 2013*), which is presumably required for pairing. Deletion of *MTH1* or *STD1* had a minimal negative effect on pairing. These results suggest that the role of Rgt1 in pairing is not simply to recruit cofactors that mediate pairing; instead, its pairing activity may compete with its transcriptional repression activity.

We hypothesized that a particular domain of Rgt1, coupled with its DNA-binding activity, might be responsible for *HAS1pr-TDA1pr* pairing. To test this idea, we generated a series of 10 amino acid

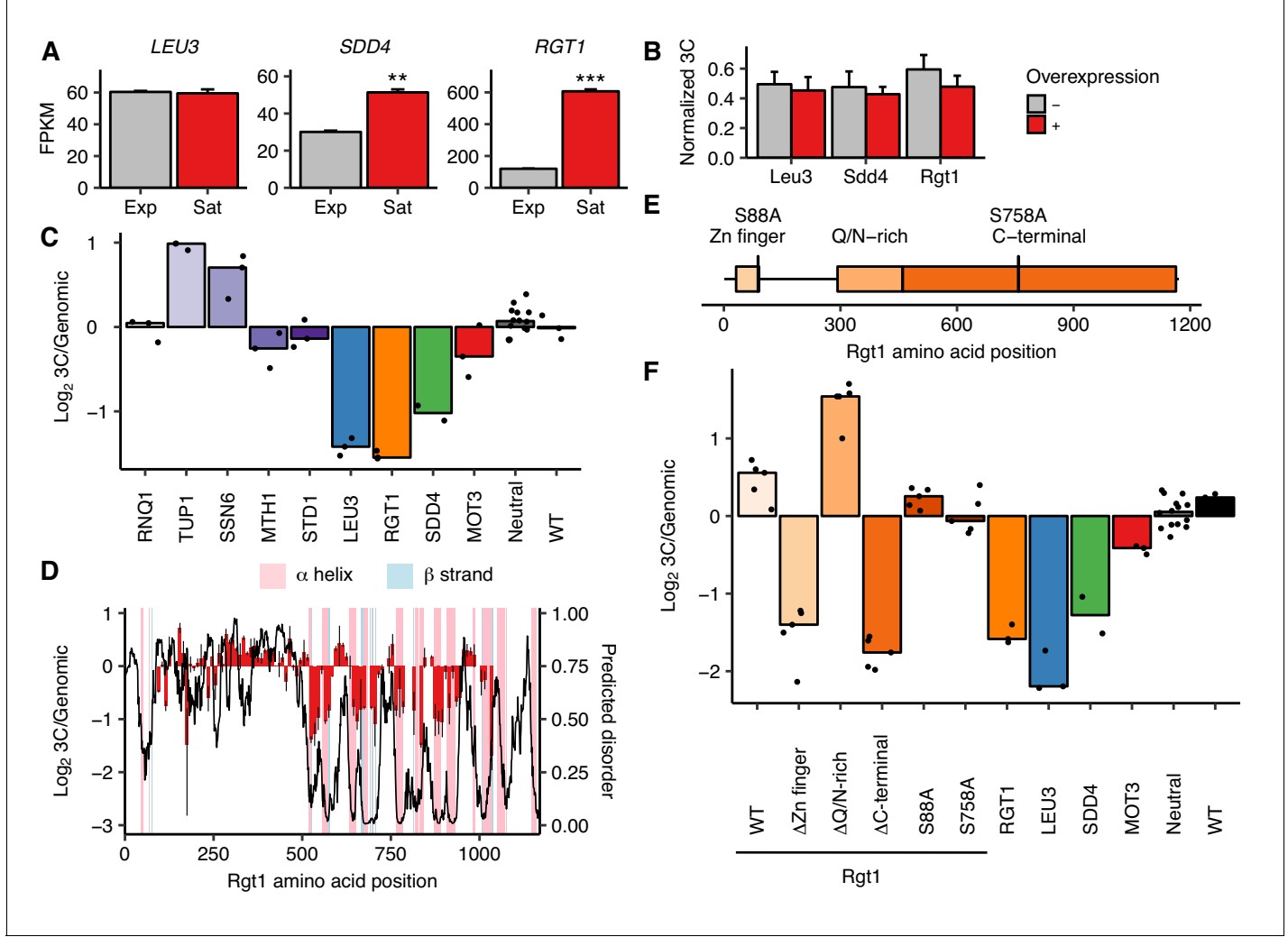

**Figure 4.** Rgt1 expression, interaction partners, and competing domains regulate *HAS1pr-TDA1pr* pairing. (**A**) *RGT1* and *SDD4* are upregulated in saturated cultures. RNA-seq expression levels for *LEU3*, *SDD4*, *RGT1* in exponentially growing (Exp) and saturated (Sat) cultures. FPKM = fragments per kilobase per million read pairs. Bars indicate mean ± s.e.m. of biological triplicates. **p<0.01, ***p<0.001 (Student's t-test). (**B**) Individual TF overexpression is insufficient for pairing. Shown are *HAS1pr-TDA1pr* homolog pairing frequencies with and without estradiol-induced overexpression of *LEU3*, *SDD4*, *RGT1* in *S. cerevisiae* x *S. uvarum* hybrids during exponential growth, measured by 3C and normalized to contacts between *HAS1pr-TDA1pr* and *LCB1* on *S. cerevisiae* chrXIII. Bars indicate mean ± s.d. of technical triplicates. (**C**) Effects of Rgt1 interaction partner deletions on ectopic *HAS1pr-TDA1pr* pairing in saturated conditions. Bar plot of median relative abundance in 3C library compared to the genomic library, with overlaid scatter plot of individual barcoded deletion strains. Retested controls are shown in same colors as in *Figure 2*; potential interaction partners are shown in shades of purple. (**D**) Regions of Rgt1 necessary for pairing in saturated culture conditions, with overlaid predicted disorder (black line, right y-axis) and secondary structure (pink and blue highlights). Each bar represents the ratio of the frequency of each 10 amino acid deletion in the 3C library compared to the genomic library. Error bars indicate the two biological replicates. (**E**) Domain deletions and phosphorylation site mutations tested in panel F. (**F**) Effects of Rgt1 domain deletions and phosphorylation site mutations on ectopic *HAS1pr-TDA1pr* pairing in saturated cultures, plotted as in panel C. Rgt1 indicates the strains with an ectopic wild-type (WT) or mutant copy of Rgt1 in addition to a deletion of the endogenous *RGT1* locus.
DOI: https://doi.org/10.7554/eLife.42499.019

The following figure supplements are available for figure 4:

**Figure supplement 1.** Overexpression of *RGT1* is not sufficient for *HAS1pr-TDA1pr* pairing.
DOI: https://doi.org/10.7554/eLife.42499.020

**Figure supplement 2.** Regions of Rgt1 required for pairing correspond to regulatory domains.
DOI: https://doi.org/10.7554/eLife.42499.021

deletions spanning most of the Rgt1 protein (positions 91–1030, excluding several deletions that dropped out during strain construction) and performed *trans* MAP-C to test the pairing function of the Rgt1 mutants (*Figure 4D*). Surprisingly, much of the C-terminal half of Rgt1 was required for *HAS1pr-TDA1pr* pairing. These 'pairing domains' are closely aligned to the regions of Rgt1 predicted to be highly structured, largely through alpha helices (*Figure 4D*). The regions of Rgt1 required for pairing also correspond loosely to those required for regulation of activation vs. repression function through allosteric changes in protein conformation in response to glucose-regulated phosphorylation (*Polish et al., 2005*) (*Figure 4—figure supplement 2*).

To further compare the roles of Rgt1 protein domains and phosphorylation in *HAS1pr-TDA1pr* pairing, we generated deletions of the Rgt1 zinc finger, Q/N-rich, and C-terminal domains, and phosphodepletion mutants S88A and S758A, and performed *trans* MAP-C to test their pairing function (*Figure 4E and F*). As expected, deletion of the zinc finger DNA-binding domain or the C-terminal domain led to background pairing levels, equivalent to lacking Rgt1 altogether. Surprisingly, deletion of the Q/N-rich domain led to stronger pairing than the wild-type control (with *RGT1* integrated at the same ectopic location), suggesting that the Q/N-rich domain inhibits pairing. Both the S88A and S758A mutations had a weak decrease in pairing, indicating that although these mutations are capable of disrupting Rgt1 activator function and intramolecular interactions (*Polish et al., 2005*), they do not individually play major roles in regulation of pairing in saturated cultures.

Together, our results suggest multiple potential modes by which the transcription factor Rgt1 regulates *HAS1pr-TDA1pr* pairing: its own expression level, recruitment of Tup1/Ssn6, and the competing activities of its Q/N-rich and C-terminal regulatory domains.

## Pairing factors do not play major roles in *HAS1* or *TDA1* transcriptional regulation

We have thus far characterized the molecular mechanism and regulation of *HAS1pr-TDA1pr* homologous pairing in saturated cultures. However, the question of why this pairing occurs (i.e. what biological function it serves, if any) remains outstanding. Now that we have identified the precise DNA base pairs and proteins involved in *HAS1pr-TDA1pr* pairing, we are equipped to test whether Leu3, Sdd4, and Rgt1 play a role in transcription at *HAS1* or *TDA1* in saturated cultures. Furthermore, we can employ *S. cerevisiae* x *S. uvarum* hybrids, in which we can discriminate the two homologous copies of each gene, in order to distinguish the effects on transcription of disrupting the pairing DNA sequence in *cis* (presumably through impaired *cis* regulation) vs. in *trans* (presumably through impaired pairing). To this end, we generated mutant hybrid strains carrying a wild-type copy of *S. uvarum HAS1pr-TDA1pr* and a copy of *S. cerevisiae HAS1pr-TDA1pr* with either a wild-type (WT) genotype, mutated Leu3 binding site (leu3), or two mutated Rgt1 sites (rgt1 $\times$ 2) (*Figure 5A*). We then performed RNA-seq in saturated culture conditions. Of note, neither of these mutants is expected to have complete loss of pairing, since the individual binding site or TF mutants disrupt pairing by ~ 4–10 fold (*Figure 1D*, *Figure 2B,C*), as opposed to the ~ 30 fold increase in pairing upon ectopic insertion of the pairing sequence (*Figure 1C*); however, if the Leu3 and Rgt1 binding site mutations cause the same effect on *HAS1* or *TDA1* transcription, the orthogonal nature of these perturbations would add confidence that any resulting molecular phenotype is an effect of disrupting pairing. The Rgt1 binding site mutations led to higher levels of *S. cerevisiae TDA1*, consistent with Rgt1 acting as a repressor under low glucose conditions (*Figure 5B*), but had no significant effect on either *S. uvarum HAS1* or *TDA1*. Surprisingly, despite the conserved binding site and strong ChIP-seq signal for Leu3 in *HAS1pr-TDA1pr*, disrupting its binding site had no significant effect on the transcript levels of any gene (*Figure 5C*).

To further characterize the transcriptional roles of Leu3, Sdd4, and Rgt1 in saturated culture, we also performed RNA-seq on deletion strains for each of these genes, along with a wild-type control, grown to saturation. As expected, each deleted gene was highly down-regulated in the mutant strains (*Figure 5—figure supplement 1A*). In addition, many known targets of Rgt1 and genes downstream of ChIP-seq peaks were highly upregulated in rgt1Δ (*Figure 5—figure supplement 2*). Focusing on the *HAS1-TDA1* locus, none of the deletion strains had altered *HAS1* expression, and only the rgt1Δ strain had even slightly altered *TDA1* expression (*Figure 5D*). These results are consistent with our earlier results from mutating TF binding sites, and together indicate that despite the importance of Leu3, Sdd4, and Rgt1 binding for pairing at the *HAS1pr-TDA1pr* locus in saturated

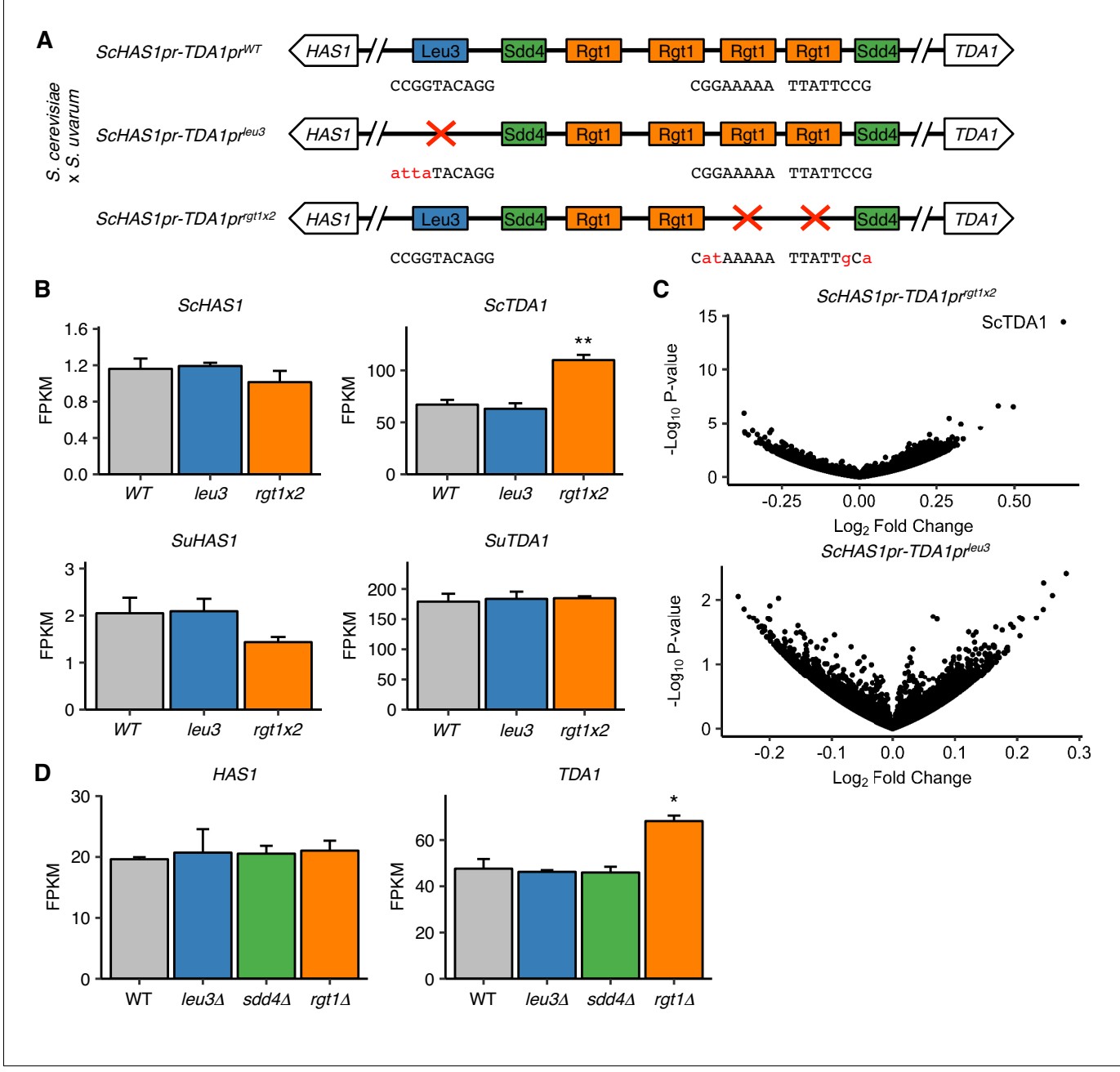

**Figure 5.** Pairing TFs play a minimal role in transcriptional regulation of the *HAS1-TDA1* locus. (**A**) Mutations in *S. cerevisiae HAS1pr-TDA1pr* tested for transcriptional effects in *S. cerevisiae* x *S. uvarum* hybrids. Colored boxes indicate TF motifs. Wild type and mutated sequences are shown below boxes, with mutated bases in lowercase, red letters. (**B**) Rgt1 regulates *TDA1* expression in cis. RNA-seq expression levels for *S. cerevisiae* (*Sc*) and *S. uvarum* (*Su*) copies of *HAS1* and *TDA1* in saturated cultures of strains shown in panel A. FPKM = fragments per kilobase per million read pairs. Bars indicate mean ± s.e.m. of biological triplicates. **p<0.01 (Student's t-test). (**C**) Mutations in pairing region cause few transcriptional changes. Volcano plot of fold change vs. p-value in mutant strains compared to wild type, shown in (**A**). (**D**) RNA-seq expression levels for *HAS1* and *TDA1* in saturated cultures of wild type haploid *S. cerevisiae* and gene deletion strains. Bars indicate mean ± s.e.m. of biological triplicates. *p<0.05 (Student's t-test).

DOI: https://doi.org/10.7554/eLife.42499.022

The following figure supplements are available for figure 5:

**Figure supplement 1.** Transcriptional effects of TF deletion in saturated culture and exponential growth.

DOI: https://doi.org/10.7554/eLife.42499.023

**Figure supplement 2.** Comparison of TF binding targets and differentially expressed genes upon TF deletion.

DOI: https://doi.org/10.7554/eLife.42499.024

culture conditions, they do not play a major role in transcriptional regulation as detected by standard polyA mRNA sequencing.

## Discussion

In summary, we developed Mutation Analysis in Pools by Chromosome conformation capture (MAP-C), a method to simultaneously test hundreds of mutations for their effects on a chromosomal contact of interest. MAP-C can be used to identify the precise sequences that are necessary for a contact (*cis* MAP-C) as well as the factors that are necessary to mediate the contact (*trans* MAP-C). Here we applied both versions of MAP-C to dissect the mechanism of inducible interchromosomal pairing between *HAS1pr-TDA1pr* alleles in budding yeast. Using a combination of gain-of-function and loss-of-function screens, we demonstrate that a trio of transcription factors—Leu3, Sdd4, and Rgt1—mediates pairing between clusters of binding sites.

Our results begin to elucidate the mechanisms of condition-specific interchromosomal contacts and homolog pairing, which have often been elusive (*Mirkin et al., 2013*). We have not yet fully defined the biochemical mechanism of pairing—it is possible the TFs interact directly and/or indirectly—but so far, no known interaction partners or cofactors have proven essential for this interaction. Unlike more prevalent nuclear-pore mediated gene relocalization and homolog pairing (*Brickner et al., 2012*; *Randise-Hinchliff and Brickner, 2016*), *HAS1pr-TDA1pr* pairing in saturated cultures does not appear to require the nuclear pore complex (*Figure 2B*). Instead, the Tup1/Ssn6 repressor complex recruited by Rgt1 negatively regulates pairing (*Figure 4C*), perhaps by inhibiting the DNA-binding activity of Rgt1 (*Roy et al., 2013*). Within Rgt1, the zinc finger DNA-binding domain and C-terminal regulatory domain are required for pairing, while the Q/N-rich region, which contains the activation domain (*Polish et al., 2005*), inhibits pairing (*Figure 4F*). These results point toward specific interactions among structured domains mediating pairing, rather than phase separation by intrinsically disordered regions.

Intriguingly, Leu3 also contains a regulatory domain whose conformation responds to environmental cues (*Sze et al., 1992*; *Wang et al., 1997*). We speculate that these regulatory domains, which are capable of intramolecular interactions (*Polish et al., 2005*), may mediate interchromosomal pairing. Furthermore, if these regulatory domains are indeed capable of direct dimerization or oligomerization, they may mediate not only interchromosomal contacts, but also cooperative TF binding at clusters of binding sites (*Kim et al., 2003*). We note that of Leu3, Sdd4, and Rgt1, only Leu3 is known to form strong dimers at each of its binding sites. However, even weak homotypic intermolecular interactions could increase the duration of interchromosomal contacts as well as increase local TF concentrations, and thereby increase DNA binding activity. It remains unclear whether heterotypic interactions among Leu3, Sdd4, and Rgt1 occur.

Another question is whether the contacts are stoichiometric, for example one-to-one, or instead mediated by aggregation of one or more TFs via weak interactions. Notably, we observe homotypic pairing between homologs of *HAS1pr-TDA1pr* and *HXT3pr* but not heterotypic pairing (*Figure 3—figure supplement 3*), suggesting that Rgt1 does not indiscriminately form contacts between all of its binding sites. However, this is not because the entire homologous context is necessary for pairing, as we found that the pairing sequence from *HAS1pr-TDA1pr* is sufficient to induce ectopic pairing in haploid *S. cerevisiae* (*Figure 1C*). Instead, a possible explanation is that only pairs of loci with a series of motifs in similar orders and orientations form frequent contacts, consistent with a stoichiometric interaction model. This added specificity beyond the presence of TF motifs could explain the lack of pairing between *HAS1pr-TDA1pr* and other sites of Leu3, Sdd4, and Rgt1 binding (*Figure 3—figure supplement 4*).

Despite our detailed molecular characterization of *HAS1pr-TDA1pr* homolog pairing, we still do not know its biological function, if any. We hypothesized that it might contribute to transcriptional regulation at either *HAS1* or *TDA1*. In particular, *TDA1* is a kinase that phosphorylates Hxk2, the main hexokinase in yeast, and thereby inhibits its ability to mediate glucose repression together with Mig1 (*Kaps et al., 2015*; *Kettner et al., 2012*) We speculated that pairing might help activate *TDA1*, which is mildly upregulated at the transcriptional level in saturated conditions (*Kim et al., 2017*). However, we find that Leu3 and Sdd4 play no detectable role in regulating transcript levels of either gene in saturated culture conditions, while Rgt1 mildly represses *TDA1* in *cis* but not in *trans* (*Figure 5*). There are several possible reasons why we may not have detected a phenotypic

effect from disrupting the pairing region or factors: (1) even low levels of pairing (~10–25% based on our MAP-C experiments shown in *Figure 1D* and *Figure 2B,C*) are sufficient to mediate its biological function; (2) its function is not in saturated culture conditions per se, but affects either exit from or re-entry into those conditions, similar to transcriptional memory at *GAL1* and *INO1* (*Brickner et al., 2015*; *Sood et al., 2017*); (3) its function only affects the dynamics of transcription (or other processes) but not steady-state RNA levels (*Zhang and Bai, 2016*). However, we also acknowledge the possibility that there is no function per se, but that pairing is instead a side-effect of conservation of biochemical features of the pairing TFs. If the same domains are required for both cooperative binding and pairing, then evolutionarily conserved pairing could result from selection on cooperative binding. Interestingly, the TF Hsf1, which forms trimers, and in flies also exhibits cooperativity between trimers (*Xiao et al., 1991*), was also recently shown to mediate interchromosomal contacts in yeast (*Chowdhary et al., 2019*). More studies of the biochemical basis of cooperative TF binding are required to test this hypothesis.

Is *HAS1pr-TDA1pr* pairing unique? Based on Hi-C, we previously found *HAS1pr-TDA1pr* to exhibit the strongest interchromosomal interactions in saturated cultures of *S. cerevisiae* x *S. uvarum* hybrids, excluding centromeres, telomeres, and the chromosomes carrying the rDNA arrays (*Kim et al., 2017*). Using ChIP-seq and motif analyses, we found that a requirement for robust adjacent DNA-binding by Leu3, Sdd4, and Rgt1 on both the *S. cerevisiae* and *S. uvarum* copies was sufficient to identify *HAS1pr-TDA1pr*. However, we also expect that the limited resolution of Hi-C and the sequence divergence in interspecific hybrids would both limit our sensitivity for detecting homolog pairing. We hypothesize that more cases of localized pairing exist, as we found with *HXT3pr* (*Figure 3D and E*). We imagine that other TFs capable of interchromosomal contacts, like Hsf1 (*Chowdhary et al., 2019*), are also capable of mediating homolog pairing, and the condition-specificity of these contacts suggests that other conditions may also exhibit similar contacts but have not yet been explored.

Although our study was focused on *HAS1pr-TDA1pr* pairing in budding yeast, both *cis* and *trans* MAP-C should be applicable to other loci and organisms. The main constraints in experimental design are that 1) the introduced mutations or associated barcodes must be included in the 3C PCR product, and 2) the region of interest should not be digested by the restriction enzyme. As we have implemented this approach, the region targeted by saturation mutagenesis was limited to < 250 bp to allow for Illumina sequencing, but this could be extended using either barcode association, similar to our *trans* knockout screens, or long-read sequencing methods. Also, we focused on a single pairwise interaction, but it is also possible to assay a mutant pool for multiple interactions, by using multiple primer pairs. MAP-C should be applicable to intrachromosomal contacts as well as interchromosomal ones, albeit with a potentially higher background of nonspecific contacts. It would be interesting to apply MAP-C to dissect the *cis* and *trans* regulators of enhancer-promoter loops in mammals, and thereby distinguish the contributions of cohesin and CTCF (*Guo et al., 2012*), general looping factors like YY1 (*Weintraub et al., 2017*), site-specific transcription factors (*Nolis et al., 2009*), and other cofactors. *Trans* MAP-C could also allow mutational scanning of TFs to clarify the biochemical mechanisms by which these transcription factors mediate chromosomal contacts.

MAP-C leverages the high throughput of saturation mutagenesis and mutant collections to allow systematic dissection of chromosome conformation. We tested up to ~ 1000 variants at a time, but with larger-scale experiments, it should be possible to test even more variants. A major potential strength of our approach is that unlike cellular high-throughput genetic screens (*Fowler and Fields, 2014*; *Gasperini et al., 2016*; *Shalem et al., 2015*), it resolves the functional consequences of mutations at the allelic level, and thus is not confounded by heterozygosity (*Patwardhan et al., 2009*). As we continue to map chromosome conformation at high resolution across ever-expanding numbers of cell types and conditions, MAP-C will provide a scalable approach to dissect the molecular mechanisms underlying specific contacts.

## Materials and methods

### Yeast strains and culture

Yeast strains used in this study are described in the Key Resources Table. Yeast were cultured at 30C, with the exception of *S. uvarum* strains, which were grown at room temperature. Cultures were

grown shaking overnight to OD$_{600}$ > 5 for saturated culture samples, or diluted to OD$_{600}$ ~ 0.125 and grown to OD$_{600}$ = 0.5–0.8 for exponential growth samples. Estradiol inductions were performed by addition of beta-estradiol to 1 μM final concentration (or equivalent volume of ethanol for negative control) to OD$_{600}$ = 0.5 cultures grown in YPD (1% w/v yeast extract, 2% w/v peptone, 2% w/v dextrose) and grown for 2 hr. Galactose induction was performed by growth in synthetic complete medium (without uracil for selection for overexpression plasmid) with 2% v/v raffinose to OD$_{600}$ = 0.75 followed by addition of 2% galactose and subsequent growth for 1.5 hr. For comparison, yeast were grown in synthetic complete medium with or without uracil with 2% glucose or 2% raffinose. *S. cerevisiae* x *S. uvarum* hybrids were generated by standard mating and auxotrophic or drug selection procedures. Yeast transformations were performed using a modified Gietz LiAc method (*Pan et al., 2004*).

## Mutant library generation

### Subsequences

All 178 bp subsequences of the intergenic region between the *S. cerevisiae HAS1* and *TDA1* coding sequences were synthesized in an Agilent (Agilent Technologies, Santa Clara, CA) array-synthesized oligonucleotide pool (sequences included in *Supplementary file 1*), and then amplified and cloned into a vector just downstream of a KanMX cassette followed by a DpnII restriction site (GATC) using NEBuilder HiFi (New England Biolabs, Ipswich, MA). These plasmids, which contain homology to the *S. cerevisiae HAS1* and *TDA1* sequences, were linearized by restriction digestion and transformed into YMD3919 (deletion of *S. cerevisiae has1pr-tda1pr*). Transformants (typically ~ 5000 per experiment) were selected on G418 medium for a total of at least 4 days (including two rounds of scraping and replating a portion onto a fresh plate), and then used to inoculate a 50 ml culture in YPD (1% w/v yeast extract, 2% w/v peptone, 2% w/v dextrose).

### Error-prone PCR

The primers TTCACCGCCTGCTATCATCC and GAATCGGCGGAATAACCTAACACG were used in a PCR reaction using the Agilent GeneMorph II kit, using as template 0.1 ng of a PCR product generated with the same primers from *S. cerevisiae* genomic DNA. The error-prone PCR products were then cloned, transformed, and selected as described above. The library contained ~ 63,000 transformants.

### Programmed 3 bp substitutions

For each set of 3 consecutive base-pairs between positions 532–675 (inclusive), we randomly chose three trinucleotide substitutions such that among them, each nucleotide change (*e.g.* A->C) was included once at each position and so that no DpnII restriction sites (GATC) were created. These sequences were synthesized on an array (sequences included in *Supplementary file 1*) and processed as described above. The library contained ~ 38,000 transformants.

### In-gene knockout screen

The selected gene knockout strains from the MATalpha yeast deletion collection (*Giaever et al., 2002*), in addition to the knockouts of the three nearest genes on either side excluding those absent or known to be slow-growing in the yeast deletion collection (*Figure 2—figure supplement 1*), were pooled and transformed *en masse* using a PCR construct with homology arms for the TEFb promoter and terminator from the KanMX deletion cassette flanking the pairing sequence (*Figure 1—figure supplement 1B*), an EcoRI restriction site, and the *URA3* selectable marker gene. The entire pool, containing ~ 12,500 transformants, was selected on synthetic complete medium without uracil and processed as described above.

### Fixed-locus knockout screens

TF genes were defined as genes with motifs on YeTFaSCo (*de Boer and Hughes, 2012*) that are not annotated as being part of a complex. Genes with no non-systematic name in the GFF file from the Saccharomyces Genome Database (SGD; version R64.2.1) were excluded. In addition, we included knockouts of genes described as nuclear pore components in SGD, the wild type strain and knockouts of eight genes known to have minimal fitness consequences under multiple nutrient limitation

conditions (*Payen et al., 2016*). The selected gene knockout strains from the MATa yeast deletion collection carrying the synthetic genetic array (SGA) reporter (*Tong et al., 2001*) (excluding those failing quality control or known to grow slowly) were grown in separate wells of deep 96-well plates, and transformed in 96-well format (*Supplementary file 3*) with a PCR construct similar to that used in the in-gene knockout screen, but with a unique 12 bp barcode added upstream of the pairing sequence (primer sequences in *Supplementary file 1*), and homology to the *YDR535C* coding sequence. Colonies were picked and verified by PCR, and one successful clone from each strain was pooled together and then diluted and grown overnight in YPD. Colonies from positive and negative control strains were repooled with new strains for subsequent fixed-locus *trans* MAP-C experiments.

## Interactor knockout screen

The selected gene knockout strains from the MATa yeast deletion collection with the SGA reporter (*Tong et al., 2001*) were each transformed with a cocktail of constructs (derived from fixed-locus screen construct) with 14 different barcodes each, and then at least eight colonies were Sanger sequenced, and three colonies from each strain carrying different barcodes were pooled and processed as described above, with positive and negative control strains included.

## Rgt1 deletion scan

The *S. cerevisiae RGT1* gene was amplified with 537 bp of upstream promoter sequence and 221 bp of downstream terminator sequence, and cloned downstream of *URA3* in the same vector used for the fixed-locus knockout screens. We attempted to create all 10 amino acid (30 bp in-frame) deletions between amino acids 91 and 1040; we successfully created 87 of the 96 attempted deletions as plasmids. The wild-type *RGT1*-containing plasmid was amplified in two pieces, each using one primer in the ampicillin resistance gene and one within *RGT1* so that the resulting pieces each contain homology for the other. These pieces were combined using NEBuilder HiFi, and then transformed into *E. coli*, extracted, and verified for the deletion junction by Sanger sequencing. Each deletion plasmid was amplified with the same primers used to create the fixed-locus knockout constructs, with a different chosen barcode for each deletion. These PCR products were pooled and transformed *en masse* into the *rgt1* deletion strain from the MATa yeast deletion collection with the SGA reporter, in two replicates. Each pool was selected on synthetic complete medium without uracil including G418 and processed as described above.

## Rgt1 domain deletion and phosphorylation site mutations

Tested mutations included deletion of the zinc finger domain (amino acids 32–90), Q/N-rich domain (amino acids 293–459), C-terminal domain (amino acids 461–1163), and S88A (TCG->GCG) and S758A (TCC->GCC). Mutant versions of the wild-type *RGT1*-containing plasmid were amplified and cloned as for the 10 amino acid deletion scan of Rgt1, and then transformed using mixtures of 14 different barcodes per desired mutation and Sanger sequenced as in the Rgt1 interactor knockout screen. Five distinct barcoded versions of each mutant were pooled and processed as above, with positive and negative control strains included.

## 3C

Cells were crosslinked by addition of 37% formaldehyde to a final concentration of 1% (v/v) and incubation at room temperature for 20 min, quenched by addition of 2.5M glycine to a final concentration of 150 mM and incubation at room temperature for 5 min, and then washed in 1x Tris-buffered saline (TBS) and stored as a pellet at −80C in aliquots of 50–100 µl dry pellets. Cells were lysed by vortexing in lysis buffer (TBS + 1% Triton X-100 with Pierce EDTA-free protease inhibitor tablet (Thermo Fisher Scientific, Waltham, MA)) with 500 µm acid-washed glass beads for 6 cycles of 2 min, with 2 min on ice between cycles. The lysate was collected by puncturing the bottom of each tube and then centrifuging the tube, stacked on top of an empty tube. The lysate was then washed in lysis buffer, then TBS, and finally resuspended in 10 mM Tris pH 8.0 to a volume of ~ 200 µl per 25 µl of starting dry pellet volume. A single 200 µl aliquot of lysate was then precleared by addition of 0.2% SDS and incubation at 65C for 10 min, cooled on ice, quenched by addition of 1% Triton X-100 (v/v), and then digested overnight with at least 100U of restriction enzyme (200U DpnII for *cis* experiments and galactose induction, 400U EcoRI-HF for *trans* experiments, and 100U NlaIII for estradiol

inductions). The restriction digest was heat-inactivated at 65C for 20 min in the presence of 1.3% SDS, and then chilled on ice and added to a dilute ligation reaction in 4 ml volume with 1% Triton X-100, 1x T4 DNA Ligase Buffer (NEB), and 10,000U of T4 DNA ligase (NEB) and incubated at room temperature for 4 hr. The ligation products were reverse-crosslinked with proteinase K at 65C overnight, and then purified by phenol-chloroform extraction followed by clean-up on a Zymo DNA Clean and Concentrator-5 column (Zymo Research, Irvine, CA). The resulting 3C DNA was quantified using a Qubit. Each technical replicate was processed separately beginning with cell lysis.

## MAP-C library preparation and sequencing

3C libraries were prepared by amplification of up to 8 reactions of 50 ng 3C DNA per replicate, using primer pairs specific to the chromosomal contact of interest (for pairing library) or a control off-target chromosomal contact (for off-target library), for 24–32 cycles. Genomic libraries were prepared by amplification of up to 4 reactions of 50 ng of either 3C DNA or genomic DNA using primers flanking the targeted mutations or barcodes, for 17–22 cycles. Reactions for each replicate were pooled, purified by Ampure XP beads (Beckman Coulter Life Sciences, Brea, CA), and then re-amplified with primers flanking the mutagenized or barcode region and including sequencing adapter sequences for 5–8 cycles, and then again with primers adding sample indices and Illumina flow-cell adapters for 6–9 cycles. All reactions were prepared with KAPA HiFi HotStart ReadyMix (Roche, Basel, Switzerland) with recommended thermocycling conditions, and included 0.5x SYBR Green I to monitor amplification by quantitative PCR and minimize the number of PCR cycles. The final libraries were sequenced on an Illumina MiSeq or Nextseq 500 (Illumina, San Diego, CA) using paired-end sequencing. See *Supplementary file 2* for detailed information on each library.

## MAP-C sequencing analysis

Paired-end reads were merged and adapter-trimmed using PEAR (*Zhang et al., 2014*), except for *trans* knockout experiments, in which only read one was used. These reads were then trimmed of the first 4 bp (corresponding to a randomized region for Illumina clustering purposes) and mapped using Bowtie 2 (*Langmead and Salzberg, 2012*) Reads were mapped to the *S. cerevisiae HAS1pr-TDA1pr* region, and then the read coverage was calculated using bedtools (*Quinlan and Hall, 2010*).

### Error-prone PCR

Reads were mapped to the wild-type sequence of the mutagenized region. The resulting alignments were scored for number of substitutions, insertions, and deletions, and the fraction of reads with a substitution at each position were calculated.

### 3 bp substitutions

Reads were mapped to the wild-type sequence of the mutagenized region. The resulting alignments were scored for number of substitutions, insertions, and deletions, and the fraction of reads with a substitution at each position were calculated, excluding reads with fewer than three substitutions (which correspond to PCR or sequencing errors of the wild-type sequence).

### Trans knockout screens, Rgt1 deletion scan, and Rgt1 mutant screen

Reads were mapped to all 192 possible barcode sequences, and the normalized fraction of reads mapping to a given barcode with a MAPQ $\geq$ 20 in the 3C library compared to the genomic control was calculated for each replicate.

## 3C qPCR

3C DNA was amplified using the same conditions as MAP-C libraries, in three replicates per primer pair. The pairing 3C products were normalized to the off-target (intrachromosomal) 3C products, assuming 2-fold amplification per cycle.

## Chromatin immunoprecipitation

Either 50 ml of exponentially growing ($OD_{600}$ = 0.7–0.8) or 12 ml of saturated cultures of TAP-tagged strains (*Ghaemmaghami et al., 2003*) were crosslinked and lysed as for 3C, but for 15 min in

FA lysis buffer (50 mM HEPES-KOH pH 7.5, 140 mM NaCl, 1 mM EDTA, 1% Triton X-100, 0.1% sodium deoxycholate, 1x Pierce EDTA-free protease inhibitor tablet). Lysates were pelleted and resuspended in FA lysis buffer, sonicated using a Diagenode Bioruptor (Diagenode, Liege, Belgium) for 3 cycles of 10 min on the HIGH power setting, with 30 s cycles on and 30 s on, and cleared by centrifugation at 20000 g for 10 min. An aliquot of 50 µl supernatant was saved for input, and the remaining sample was incubated with 10 µl of Dynabeads Pan Mouse IgG magnetic beads (pre-washed with FA lysis buffer) rotating overnight at 4C. The immunoprecipitations were washed twice with FA lysis buffer, once with high salt (500 mM NaCl) FA lysis buffer, twice with RIPA buffer (10 mM Tris-HCl pH 8, 250 mM LiCl, 0.5% Igepal CA-630, 0.5% sodium deoxycholate, 1 mM EDTA), and once with TE (50 mM Tris-HCl pH 8, 1 mM EDTA), and then eluted first with 100 µl TE + 1% SDS at 65C for 15 min, and a second time with 150 µl TE + 0.67% SDS. Inputs were diluted with 200 µl of TE + 1% SDS, and all samples were treated with 50 mg RNase A at 37C for 10 min and 100 mg proteinase K at 42C for 1 hr, and then reverse crosslinked overnight at 65C. DNA was purified with a Zymo ChIP DNA Clean and Concentrator-5 kit and eluted in 15 µl of 10 mM Tris-HCl pH 8.

qPCRs were performed using 5 µl of either 1:20 dilution of IP samples or 1:800 dilution of input samples, in 25 µl reactions with KAPA Robust 2G HotStart ReadyMix, using 0.5x SYBR Green I and standard cycling conditions for 40 cycles on a Bio-Rad C1000 Touch thermal cycler with a CFX96 Real-Time System (Bio-Rad Laboratories, Hercules, CA). Cq values were calculated using the Bio-Rad CFX Manager 3.1 software, using the single threshold mode for Cq calculation and baseline subtracted curve fitting. Primer efficiencies were calculated using a 5-fold dilution series of genomic DNA from *S. cerevisiae* BY4741 starting from 20 ng. Primer sequences are included in *Supplementary file 1*.

ChIP-seq libraries were prepared using Swift Accel-NGS 2S Plus (Swift Biosciences, Ann Arbor, MI) dual-indexed kits using 10 µl of IP samples or 1 ng of input samples, with 9 cycles of PCR for input samples and 12–15 cycles for IP samples. Libraries were sequenced to ~ 2.5–6 million read pairs per sample using 2 × 37 bp reads on an Illumina NextSeq 500.

## ChIP-seq analysis

Sequencing reads were first pre-processed using cutadapt (*Martin, 2011*): reads were quality-trimmed (option -q 20), trimmed of adapter sequences, excluding any read pairs in which either read was shorter than 28 bp after trimming (option -m 28). Read pairs were mapped to the sacCer3 *S. cerevisiae* reference genome using Bowtie 2 (*Langmead and Salzberg, 2012*) with the –very-sensitive parameter set, requiring the reads in each pair to be with 2000 bp of each other (option -X 2000). Read pairs in which both reads had a mapping quality score of at least 30 were deduplicated using samtools rmdup. Replicates were merged prior to calling peaks and generating fold enrichment tracks using MACS2 (https://github.com/taoliu/MACS) (*Zhang et al., 2008*). Fold enrichment tracks were visualized using the UCSC Genome Browser (*Karolchik et al., 2003*).

## RNA sequencing

Yeast strains were grown overnight in YPD in biological triplicate (independent colonies, or for newly generated transformants, independent transformants), pelleted and then stored at −80C. RNA was purified using acid phenol extraction, and then treated with Turbo DNase and purified with a Qiagen RNeasy Mini kit (Qiagen, Hilden, Germany). Illumina libraries were prepared from 800 to 900 ng of total RNA, using the Illumina Truseq RNA Library Prep kit v2 (for TF binding site mutants) or the Illumina Truseq Stranded mRNA Library Prep kit (for TF knockouts). Libraries were sequenced to ~ 11–13 million read pairs per sample using 2 × 37 bp reads on an Illumina NextSeq 500.

## RNA-seq analysis

Reads were preprocessed and mapped as with the ChIP-seq libraries, but with the -X 500 option for Bowtie 2. Read pairs in which both reads had a mapping quality score of at least 30 were overlapped with annotated genes using HTSeq (*Anders et al., 2015*). Global fold-change analyses were using DESeq2 (*Love et al., 2014*).

## RT-qPCR

Yeast were grown as described above in Yeast strains and culture, and then pelleted and stored at −80C. RNA was purified using acid phenol extraction, and then treated with Turbo DNase and purified with a Qiagen RNeasy Mini kit. For each sample, 1 ug of total RNA was annealed to oligo(dT)$_{20}$ (13 μl reaction with 1 μl of 50 μM oligo(dT), 1 μl 10 mM dNTP mix) by incubating at 65C for 5 min and then on ice for 1 min. Reverse transcription was performed with SuperScript IV by adding 4 μl of 5x SuperScript IV buffer, 1 μl SUPERase In RNase inhibitor, 1 μl 100 mM DTT, and 1 μl of SuperScript IV enzyme, and then incubating at 50C for 10 min and then 80C for 10 min. For qPCRs, 2.5 μl of each reverse transcription reaction was used for each 25 μl PCR, using KAPA Robust 2G HotStart ReadyMix with standard cycling conditions (except annealed at 55C) for 40 cycles on a Bio-Rad C1000 Touch thermal cycler with a CFX96 Real-Time System. Cq values were calculated using the Bio-Rad CFX Manager 3.1 software, using the regression mode for Cq calculation and baseline subtracted curve fitting. Primer efficiencies were calculated using a 5-fold dilution series of genomic DNA from *S. cerevisiae* BY4741 starting from 10 ng. Primers were specific to *S. cerevisiae* (at least two substitutions to *S. uvarum*). See *Supplementary file 1* for primer sequences.

## Rgt1 protein annotations

Predicted intrinsic disorder was calculated using IUPred2 long disorder (*Mészáros et al., 2018*). Predicted secondary structure was calculated using Jpred4 (*Drozdetskiy et al., 2015*) separately on the first 400 amino acids and on the remaining 770 amino acids, as submissions are capped at 800 amino acids.

## Motif analysis

Systematic scans of motifs were performed using YeTFaSCo (*de Boer and Hughes, 2012*) with the expert-curated no dubious motif set. Sequence logos were generated using ggseqlogo (*Wagih, 2017*).

## Motif cluster analysis

Motif clusters were identified using MCAST (*Grant et al., 2016*) with a motif p-value threshold of 0.001, a maximum gap threshold of 200 bp, and an E-value threshold of 2, using the high-confidence motifs for Leu3 (#781), Rgt1 (#2227), and Sdd4 (#588) from YeTFaSCo (*de Boer and Hughes, 2012*). Individual motif occurrences for each TF were scored for the *S. cerevisiae* (version R64.2.1) and *S. uvarum* genomes (as revised in *Kim et al., 2017*) using FIMO (*Grant et al., 2011*) with a p-value threshold of 0.001 and the option —max-strand. Motif clusters were named based on the nearest downstream gene (on - strand if coordinate of gene < coordinate of motif cluster, and on + strand if coordinate of gene > coordinate of motif cluster). Motif clusters were defined to be homologous if they were upstream of homologous genes.

## De novo motif discovery

De novo motif discovery was performed using MEME version 4.12.0 (*Bailey et al., 2006*). For all analyses, 100 bp centered at each ChIP-seq peak or tRNA gene was used. Motifs were allowed to be between 6 bp and either 10 bp or 20 bp, and the top three motifs were analyzed, with otherwise default settings.

## Hi-C

Hi-C was performed and analyzed as in *Kim et al. (2017)* using the restriction enzyme Sau3AI.

## Code availability

Code used to analyze data and generate figures are available at https://github.com/shendurelab/MAP-C (*Kim, 2019*; copy archived at https://github.com/elifesciences-publications/MAP-C).

## Data availability

All sequencing data have been deposited in the Gene Expression Omnibus (GEO) under accession number GSE118118. Hi-C data from *Figure 3* and *Figure 3—figure supplements 2*, *3* and *4* and RNA-seq data from *Figure 4* are from GEO accession number GSE88952. Processed microarray

data of gene expression in TF deletions under exponential growth from *Figure 5—figure supplements 1* and *2* are from GEO accession number GSE4654 (*Hu et al., 2007*).

## Acknowledgements

We thank K Xue, W Noble, and members of the Shendure and Dunham labs, particularly V Agarwal, J Tome, and S Domcke, for comments and discussion; I Liachko for help with Hi-C experiments; N Hanson for help with yeast deletion collections; and Y Zheng, M Sanchez, and RS McIsaac for strains.

## Additional information

### Funding

| Funder | Grant reference number | Author |
|---|---|---|
| National Institutes of Health | U54 DK107979 | Jay Shendure |
| National Science Foundation | Graduate Research Fellowship DGE-1256082 | Seungsoo Kim |
| Howard Hughes Medical Institute | Investigator | Jay Shendure |
| Canadian Institute for Advanced Research | Senior Fellow (Genetic Networks Program) | Maitreya J Dunham |

The funders had no role in study design, data collection and interpretation, or the decision to submit the work for publication.

### Author contributions

Seungsoo Kim, Conceptualization, Investigation, Visualization, Methodology, Writing—original draft, Writing—review and editing; Maitreya J Dunham, Jay Shendure, Supervision, Writing—review and editing

### Author ORCIDs

Seungsoo Kim  https://orcid.org/0000-0002-5559-5289
Maitreya J Dunham  http://orcid.org/0000-0001-9944-2666
Jay Shendure  https://orcid.org/0000-0002-1516-1865

### Decision letter and Author response

Decision letter https://doi.org/10.7554/eLife.42499.037
Author response https://doi.org/10.7554/eLife.42499.038

## Additional files

### Supplementary files

• Supplementary file 1. Oligonucleotide sequences used in this study.
DOI: https://doi.org/10.7554/eLife.42499.025

• Supplementary file 2. MAP-C libraries.
DOI: https://doi.org/10.7554/eLife.42499.026

• Supplementary file 3. 96-well yeast transformation protocol.
DOI: https://doi.org/10.7554/eLife.42499.027

• Supplementary file 4. Key resources table.
DOI: https://doi.org/10.7554/eLife.42499.028

• Transparent reporting form
DOI: https://doi.org/10.7554/eLife.42499.029

### Data availability

All sequencing data have been deposited in GEO under accession code GSE118118.

The following dataset was generated:

| Author(s) | Year | Dataset title | Dataset URL | Database and Identifier |
|---|---|---|---|---|
| Kim S, Dunham MJ, Shendure J | 2018 | A combination of transcription factors mediates inducible interchromosomal contacts | https://www.ncbi.nlm.nih.gov/geo/query/acc.cgi?acc=GSE118118 | NCBI Gene Expression Omnibus, GSE118118 |

The following previously published datasets were used:

| Author(s) | Year | Dataset title | Dataset URL | Database and Identifier |
|---|---|---|---|---|
| Kim S, Liachko I, Brickner DG, Cook K, Noble WS, Brickner JH, Shendure J, Dunham MJ | 2017 | The dynamic three-dimensional organization of the diploid yeast genome | https://www.ncbi.nlm.nih.gov/geo/query/acc.cgi?acc=GSE88952 | NCBI Gene Expression Omnibus, GSE88952 |
| Hu Z, Killion PJ, Iyer VR | 2007 | Genetic reconstruction of a functional transcriptional regulatory network | https://www.ncbi.nlm.nih.gov/geo/query/acc.cgi?acc=GSE4654 | NCBI Gene Expression Omnibus, GSE4654 |

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
