## [Decision Letter]

Thank you for submitting your article "A combination of transcription factors mediates inducible interchromosomal contacts" for consideration by *eLife*. Your article has been reviewed by has been reviewed by three peer reviewers, including Jeannie Lee as the Reviewing Editor and Reviewer #3, and the evaluation has been overseen by a Senior Editor. The following individuals involved in review of your submission have agreed to reveal their identity: Peter J Fraser (Reviewer #1); Hsueh ping Chu (Reviewer #2).

We find your study to be well-conducted and the findings to be broadly interesting to the community.

Summary:

The authors present the development of MAP-C (Mutation Analysis in Pools by Chromosome conformation capture) to study the regulation of chromosomal interactions through mutagenesis. Two flavors of the methods allow (i) identifying the local sequence required for interaction by cis MAP-C (ii) testing the requirement of other factors or sequences for interaction by trans MAP-C. The authors demonstrate the power of the method by applying it to the two HAS1pr-TDA1pr alleles, which they recently demonstrated to pair under defined biological conditions. The authors went on to identify the local genomic sequence and three transcription factors essential for allelic pairing of HAS1pr-TDA1pr. The authors suggest that a regulated interaction between the three factors controls both the specificity and physiological regulation of HAS1pr-TDA1pr pairing. The tools described will facilitate the study of pairing, an emerging epigenetic phenomenon that is still poorly understood.

Essential revisions:

Please focus on several of the key points raised by reviewers:

1) The identification of 3 TFs for pairing suggests that transcriptional activity or RNA produced at the locus might play a role in the trans interaction. The authors should characterize the transcriptional status of the locus and its changes in the individual TFs knockouts and upon pairing.

2) While the C-terminal region of Rgt1 seems required for pairing, the suggestion that it is regulated by phosphorylation is a stretch. We recommend testing the effect of mutating the regulatory S88 and S758 residues.

3) Reviewer 1 would like to see either data or discussion of the biological purpose of localized pairing and why it is inducible. This is a fair question and I (reviewer 3) am also interested in the thinking behind it.

4) We would require evidence that the factors actually bind the pairing locus. ChIP-seq of these three transcription factors to verify the binding of Leu3, Sdd4 and Rgt1 at the minimal pairing region would strengthen the paper.

5) Data to address the differences in binding level of the three TFs at the minimal pairing region under the two different growth conditions. ChIP-seq is again recommended.

6) As mentioned by reviewers below, there are other major cases of pairing in various organisms, including mammals. In the revision, please be sure to properly cite those studies to put the current study in perspective.

7) Reviewer 2 also asked if homologous pairing at HAS1pr-TDA1pr regulates nearby gene expression in saturated cultures. If data are available to address this, please include them.

*Reviewer #1:*

This research is a follow-up study of this group's previous studies published in *eLife* (Kim et al., 2017). In their previous publication, this group used diverged haploid yeast as a model (*S. cerevisiae* ×*S. uvarum*) and provided a global view of chromosome conformation in hybrid diploid yeasts. They found localized increases in homologous interactions between the HAS1-TAD1 alleles specifically under galactose induction and saturated growth conditions.

The current study aims to understand the regulatory mechanism underlying homologous pairing between the HAS1-TAD1 locus in diploid yeast. To identify the specific sequence elements that are necessary and sufficient to maintain homolog pairing between the HAS1-TAD1 alleles, the authors developed a method called MAP-C (Mutation Analysis in Pools by Chromosome conformation capture). The cis MAP-C method involves analyzing populations of cells with multiple mutations of specific regions suspected to mediate interchromosomal pairing followed by amplification of 3C products and sequencing. To find the minimal subsequence of the 1,038 bp HAS1pr-TDA1pr region that is essential for the homologous interactions, the authors replaced the native *S. cerevisiae* HAS1pr-TDA1pr locus with a library containing each of 861 tiling 178 bp subsequences of the 1,038 bp region. Following identification of the minimal pairing region, they used error-prone PCR to obtain a base-pair resolution map of the DNA sequences necessary for homologous pairing, and identified unique clusters of sequence motifs in the minimal pairing region, which correspond to specific TFs binding motifs. Next, they used trans MAP-C, to test trans knockouts of the TF candidates identified in the cis MAP-C and confirmed that three TFs (Leu3, Sdd4, and RGT1) mediate the interchromosomal pairing. The method variants are interesting and useful, however there are a few points that could be addressed that could potentially make the manuscript more interesting for the readership of *eLife*.

1) Obviously, the homologous pairing between HAS1-TAD1 alleles is a localized event, and the pairing is inducible (growth saturation or galactose induction) and limited into a 1kb intragenic regions on chromosome XIII. However, this study did not attempt to answer key biological questions: what's the biological purpose or meaning of this localized pairing? Why it's inducible? There are no experimental data or discussions in the manuscript. This is an important question that could greatly improve the interest level and relevance of the findings and the manuscript.

2) In the Introduction, the author described that "a motif match is insufficient to fully predict TF binding", which is true. Later on, this study identified binding of the three transcription factors binding to the "minimal pairing region" which are necessary and essential for homologous pairing between HAS1-TAD1 alleles. However, most of the Results section and Conclusion were based on motif scanning, motif searching, and mutations in this region. This study missed some very important experiments: ChIP-seq of these three transcription factors to verify the binding of Leu3, Sdd4 and Rgt1 at the minimal pairing region.

3) Since the homologous pairing between HAS1-TAD1 alleles is significantly higher at growth saturation than at standard growth conditions, and this study also identified that three transcription factors (Leu3, Sdd4 and Rgt1) regulate this homolog pairing. What's the binding level of these three TFs at the minimal pairing region under the two different growth conditions? ChIP-seq of these three transcription factors under the two different growth conditions is very important to verify and solidify the conclusion.

4) The authors claimed that HAS1pr-TDA1pr pairing is regulated by the expression level of TFs, particularly RGT1. In fact, using the mRNA expression level of TFs is not convincing way to convey this idea. Because increasing at mRNA level does not necessarily means increasing TFs' bindings at chromatin, DNA, or a specific loci (in the current study). The best experiment here would again be ChIP-seq of the three TFs to show differential binding.

5) Last part of the Results section, there is no data to support that the "pairing domains" of Rgt1 needs to be phosphorylated for the purpose of homologous pairing. This is speculation without supporting data. If the authors believe that phosphorylation of the "pairing domains" of Rgt1 is one of the regulatory mechanisms of Rgt1 function, then they need to show that under growth saturation conditions, the "pairing domain" of Rgt1 is indeed phosphorylated.

6) In the Introduction the authors claim that:

"Although interchromosomal contacts, such as those between actively transcribed genes or homolog pairing in mitotically dividing yeast, have been observed by microscopy (Burgess et al., 1999; Lim et al., 2018; Maass et al., 2018), many are not detectable using 3C (chromosome conformation capture) technologies. Some have argued that these contacts might be a result of fluorescent in situ hybridization (FISH) artifacts (Lorenz et al., 2003) or the insertion of binding site arrays often used for live-cell imaging (Mirkin et al., 2014). A more detailed understanding of the molecular components of interchromosomal contacts is needed to resolve this ambiguity."

Although this passage is written to suggest they are only referring to yeast, the papers cited in fact refer to *Drosophila* and mouse cells. There are in fact many publications that have shown a correspondence between interchromosomal associations between transcribed genes by FISH and by 3C in mammalian cells going back over a decade, and more recently in live yeast cells with 3C technology verification. As always authors like to suggest novelty in order to get published, but there is no need to ignore the work of many other colleagues and many other publications. The author's method alone and its findings are powerful results. The authors should read and cite those original papers instead of making an erroneous claim that will only serve to irritate their colleagues.

*Reviewer #2:*

In the manuscript, the authors developed a method to study interchromosomal interactions to simultaneously analyze hundreds of cis or trans-acting mutations. This method will advance the dissection of the molecular mechanisms of the chromosomal pairing. The authors identified the minimal sequence that is essential for the homologous pairing at the HAS1pr-TDA1pr locus. The pairing event occurs particularly in the saturated cultures in budding yeast. They introduced a barcode to each gene knockout and tested hundreds of transcription factors knockouts. The authors found that LEU3, SDD4, and RGT1 are required for pairing. Finally, they identified a particular domain of RGT1 that is responsible for the pairing. This study demonstrated that transcription factors play an important role in chromosomal interactions using genetic approaches. In my opinion, this study was well done.

1) The authors analyze the DNA binding sites for RGT1, LEU3, and SDD4 at the HAS1pr-TDA1pr locus (Figure 1.E-H). However, there is no evidence showing that these transcription factors directly bind to HAS1pr-TDA1pr locus in saturation cultures but not in exponentially growing cultures. Chromatin immunoprecipitation for RGT1 and LEU3 could answer this question.

2) In Figure 3C legend), there is a typo in the sentence: {greater than or equal to}2 bins from a telomere. What does "bin" mean in the figure legend?

3) Does the homologous pairing at HAS1pr-TDA1pr regulate nearby gene expression in saturated cultures? The authors could analyze the RNA-seq data to address this question.

*Reviewer #3:*

Kim et al., present the development of MAP-C (Mutation Analysis in Pools by Chromosome conformation capture), a new approach to study the regulation of chromosomal interactions by mutation. Two flavors of the methods allow (i) identifying the local sequence required for interaction by cis MAP-C (ii) testing the requirement of other factors or sequences for interaction by trans MAP-C. The authors demonstrate the power of the method by applying it to the two HAS1pr-TDA1pr alleles, which they recently demonstrated to pair under defined biological conditions.

They identify the local genomic sequence and three transcriptions factors, which they demonstrate to be essential for allelic pairing of HAS1pr-TDA1pr. The authors suggest that a regulated interaction between the three factors controls both the specificity and physiological regulation of HAS1pr-TDA1pr pairing.

Kim et al., describe a method allowing to probe the molecular mechanism of chromosomal pairing. They identify both the precise genomic sequence and the factors required for inter-chromosomal interaction. In the future, this new tool will facilitate the study of pairing, a widely observed but still poorly characterized biological enigma.

- Subsection “A cluster of TF motifs is necessary and sufficient for *HAS1pr-TDA1pr* pairing” and Figure 1D: The authors claim to identify five clusters, however this choice seems arbitrary with additional depletions of substituted nucleotides (e.g regions upstream of E and between G-H).

- Figure 1—figure supplement 2C: While the intensity of ratios is clearly less intense in the control than in the 3C library, it shows a similar pattern (compare panel C to panel B).

- Subsection “Three transcription factors are required for pairing”, second paragraph is difficult to understand. How does the position of deleted regions influence the interactions? This should be improved together with the explanation of trans MAP-C and Figure 2A.

- The identification of three transcriptional factors (TFs) requird for HAS1pr-TDA1pr pairing suggests that transcriptional activity or RNA produced at the locus might play a role in regulating the interaction between the two regions. The authors should characterize the transcriptional status of the locus and its changes in the individual TFs knockouts and upon pairing.

- Rgt1 domain analysis showed that the C-terminal region, largely structured, is required for pairing. However, there is quite a stretch from this observation to the suggestion that it is regulated by phosphorylation, as previously described (Polish et al., 2005). The authors should test the effect of mutating the regulatory S88 and S758 residues.

[Editors' note: further revisions were requested prior to acceptance, as described below.]

Thank you for resubmitting your work entitled "A combination of transcription factors mediates inducible interchromosomal contacts" for further consideration at *eLife*. We apologize for the additional time it has taken to render this decision. This is due to the fact that we waited for the remaining reviewer to respond. Although we never received their comments, we have decided to proceed on the basis of one review. Your revised article has been favorably evaluated by us and also by one of the original reviewers.

The manuscript has been improved but there is one remaining issue that needs to be addressed before acceptance. Reviewer 2 requests evidence that pairing has been successfully disrupted in order to validate interpretation of the results. Specifically, mutating Rgt1 binding sites results in a TDA1 upregulation but mutating Leu3 binding sites does not (Figure 5). It is important to determine by 3C-qPCR (for example) that pairing is lost in the mutants. The Log23C/genome data shown is not sufficient to make the claim. Reviewer 2 suggested that the authors compare HAS1pr-TDA1pr pairing with a non-pairing region or under exponential growth conditions. We believe that this experiment should not take too long to perform and would enable us to move forward with the manuscript, assuming all goes as planned.

*Reviewer #2:*

The revised manuscript includes ChIP-seq and RNA-seq data to investigate the mechanism of HAS1pr-TDA1pr pairing. The authors show that Leu3, Sdd4, and Rgt1 were more enriched at HAS1pr-TDA1pr locus in saturated culture conditions than in exponential growth condition, suggesting that these transcription factors are directly involved in the pairing. They performed RNA-seq for those TF mutants and TF binding site mutants and found that those mutations have little effects on transcription levels of HAS1 and TDA1 genes, and they haven't found the biological functions of the pairing in their manuscript. Several possible reasons why they have not detected a phenotypic effect from disrupting the pairing region or factors were discussed in the revised version. Although they have mostly addressed the questions I have, there are few concerns about whether they have successfully disrupted the pairing that may lead to a different interpretation of the results.

The authors found that the mutation of Rgt1 binding sites at the pairing locus results in a mild upregulation of TDA1 expression (Figure 5B), but the mutation of Leu3 binding sites does not affect on gene expression (Figure 5C). The authors claim that the disruption of pairing may not have any effect on transcription (subsection “Pairing factors do not play major roles in *HAS1* or *TDA1* transcriptional regulation”). However, I don't see the measurement of the pairing events (such as 3C-qPCR) for those mutants (Figure 5A). Do these mutants completely disrupt the pairing? They used Log23C/genome to show the decrease of pairing in those mutants, but it doesn't represent the level of pairing remained in those mutants. The authors should compare HAS1pr-TDA1pr pairing with a non-pairing region (or in exponential growth condition) as a control to show whether those mutants have entirely abolished the pairing or not.

---

## [Author Response]

[…] Essential revisions:Please focus on several of the key points raised by reviewers:1) The identification of 3 TFs for pairing suggests that transcriptional activity or RNA produced at the locus might play a role in the trans interaction. The authors should characterize the transcriptional status of the locus and its changes in the individual TFs knockouts and upon pairing.

In response to this comment, we performed RNA sequencing of both TF binding site mutants in *S. cerevisiae* x *S. uvarum* hybrids, in order to detect potential *trans* effects that might result from disruption of pairing, as well as of TF knockouts in *S. cerevisiae*, all in the saturated culture conditions that produce *HAS1pr-TDA1pr* pairing. In addition, we analyzed existing microarray data of the same TF deletions in exponentially growing cells (GSE4654, Hu et al., 2007). As shown in Figure 5, we find that either mutating two Rgt1 binding sites in the *HAS1pr-TDA1pr* region or deleting the *RGT1* gene result in mild (~1.5-2-fold) upregulation of *TDA1*, consistent with its known role as a repressor in low glucose conditions. However, these effects were limited to the copy of *TDA1* adjacent to the mutations and indicate that pairing is not necessary for normal regulation of *TDA1* under saturated conditions. Furthermore, none of the other perturbations had any significant effect on *HAS1* or *TDA1* expression. In the Discussion section, we describe the implications and possible reasons for a lack of a detected function for pairing.

2) While the C-terminal region of Rgt1 seems required for pairing, the suggestion that it is regulated by phosphorylation is a stretch. We recommend testing the effect of mutating the regulatory S88 and S758 residues.

We have performed a MAP-C experiment with the Rgt1 S88A and S758A mutations described in Polish et al., 2005, alongside the ectopic wild-type Rgt1 control and deletions for the zinc finger, Q/N-rich, and C-terminal regulatory domains. As shown in Figure 4E and 4F, we do not find strong effects of preventing phosphorylation, only mild decreases in pairing. However, we suspect that Rgt1 would predominantly be in a dephosphorylated state under saturated culture conditions, and thus, our results do not rule out a role for phosphorylation in regulating pairing activity, i.e. preventing pairing in exponential growth conditions. But in light of these results, we have de-emphasized the possible role of phosphorylation in regulating pairing.

3) Reviewer 1 would like to see either data or discussion of the biological purpose of localized pairing and why it is inducible. This is a fair question and I (reviewer 3) am also interested in the thinking behind it.

We have added extensive discussion of the function of *HAS1pr-TDA1pr* homolog pairing, or rather, why we fail to detect a molecular phenotype of disrupting either pairing TFs or their binding sites in the pairing region. As described in the revised Discussion section, there are various possible reasons why we have not detected a function: (1) even weak pairing (such as that without all three TFs and their binding sites) may suffice for its function; (2) pairing function may not act during saturated culture conditions, but instead upon exit from or re-entry into those conditions, as in epigenetic transcriptional memory; (3) pairing may regulate transcriptional dynamics rather than steady-state transcript levels. Finally, we speculate regarding the possibility that pairing has no function per se but is instead the result of conservation of homotypic intermolecular interactions among TFs that also mediate cooperative DNA binding.

4) We would require evidence that the factors actually bind the pairing locus. ChIP-seq of these three transcription factors to verify the binding of Leu3, Sdd4 and Rgt1 at the minimal pairing region would strengthen the paper.

We have performed ChIP-qPCR and ChIP-seq for Leu3, Sdd4, and Rgt1 using their TAP-tagged versions, both in haploid *S. cerevisiae* grown to saturation and in exponentially growing cultures. As shown in Figure 2D and E, all three TFs bind much more strongly in saturated cultures, with only Leu3 showing detectable binding in exponential growth. We further use these data to extend our motif analyses assessing the uniqueness of *HAS1pr-TDA1pr* pairing and whether combinatorial TF binding specifies where homolog pairing occurs in saturated culture conditions (shown in Figure 3). We find that *HAS1pr-TDA1pr* indeed exhibits the strongest coincident binding of all three TFs and conserved motif clusters in both *S. cerevisiae* and *S. uvarum* genomes. However, such coincident binding does not appear to be necessary for detectable inducible homolog pairing, as the copies of the *HXT3* promoter exhibit pairing without Leu3 binding and only weak Sdd4 binding (Figure 3A).

5) Data to address the differences in binding level of the three TFs at the minimal pairing region under the two different growth conditions. ChIP-seq is again recommended.

Please see our response to the last point.

6) As mentioned by reviewers below, there are other major cases of pairing in various organisms, including mammals. In the revision, please be sure to properly cite those studies to put the current study in perspective.

We would like to thank the reviewers for suggesting relevant literature. We have added discussion of other known cases of homolog pairing, including genome-wide pairing in flies and X chromosome pairing in mice, to the Introduction.

7) Reviewer 2 also asked if homologous pairing at HAS1pr-TDA1pr regulates nearby gene expression in saturated cultures. If data are available to address this, please include them.

Please see our response to point #1 above. We have also previously found (Kim et al., 2017) that *HAS1* is strongly downregulated in saturated culture conditions, and *TDA1* is mildly upregulated (compared to exponential growth) and mention the expression of *TDA1* in the Discussion section.

Reviewer #1:[…] 1) Obviously, the homologous pairing between HAS1-TAD1 alleles is a localized event, and the pairing is inducible (growth saturation or galactose induction) and limited into a 1kb intragenic regions on chromosome XIII. However, this study did not attempt to answer key biological questions: what's the biological purpose or meaning of this localized pairing? Why it's inducible? There are no experimental data or discussions in the manuscript. This is an important question that could greatly improve the interest level and relevance of the findings and the manuscript.

Please see our responses to main points #1, #3, and #7.

2) In the Introduction, the author described that "a motif match is insufficient to fully predict TF binding", which is true. Later on this study identified binding of the three transcription factors binding to the "minimal pairing region" which are necessary and essential for homologous pairing between HAS1-TAD1 alleles. However, most of the Results section and Conclusion were based on motif scanning, motif searching, and mutations in this region. This study missed some very important experiments: ChIP-seq of these three transcription factors to verify the binding of Leu3, Sdd4 and Rgt1 at the minimal pairing region.

Please see our responses to main points #4 and #5.

3) Since the homologous pairing between HAS1-TAD1 alleles is significantly higher at growth saturation than at standard growth conditions, and this study also identified that three transcription factors (Leu3, Sdd4 and Rgt1) regulate this homolog pairing. What's the binding level of these three TFs at the minimal pairing region under the two different growth conditions? ChIP-seq of these three transcription factors under the two different growth conditions is very important to verify and solidify the conclusion.

Please see our responses to main points #4 and #5.

4) The authors claimed that HAS1pr-TDA1pr pairing is regulated by the expression level of TFs, particularly RGT1. In fact, using the mRNA expression level of TFs is not convincing way to convey this idea. Because increasing at mRNA level does not necessarily means increasing TFs' bindings at chromatin, DNA, or a specific loci (in the current study). The best experiment here would again be ChIP-seq of the three TFs to show differential binding.

Please see our responses to main points #4 and #5.

5) Last part of the Results section, there is no data to support that the "pairing domains" of Rgt1 needs to be phosphorylated for the purpose of homologous pairing. This is speculation without supporting data. If the authors believe that phosphorylation of the "pairing domains" of Rgt1 is one of the regulatory mechanisms of Rgt1 function, then they need to show that under growth saturation conditions, the "pairing domain" of Rgt1 is indeed phosphorylated.

Please see our responses to main point #2.

6) In the Introduction the authors claim that:"Although interchromosomal contacts, such as those between actively transcribed genes or homolog pairing in mitotically dividing yeast, have been observed by microscopy (Burgess et al., 1999; Lim et al., 2018; Maass et al., 2018), many are not detectable using 3C (chromosome conformation capture) technologies. Some have argued that these contacts might be a result of fluorescent in situ hybridization (FISH) artifacts (Lorenz et al., 2003) or the insertion of binding site arrays often used for live-cell imaging (Mirkin et al., 2014). A more detailed understanding of the molecular components of interchromosomal contacts is needed to resolve this ambiguity."Although this passage is written to suggest they are only referring to yeast, the papers cited in fact refer to Drosophila and mouse cells. […] The authors should read and cite those original papers instead of making an erroneous claim that will only serve to irritate their colleagues.

Please see our responses to main point #6.

Reviewer #2:.[…] 1) The authors analyze the DNA binding sites for RGT1, LEU3, and SDD4 at the HAS1pr-TDA1pr locus (Figure 1.E-H). However, there is no evidence showing that these transcription factors directly bind to HAS1pr-TDA1pr locus in saturation cultures but not in exponentially growing cultures. Chromatin immunoprecipitation for RGT1 and LEU3 could answer this question.

Please see our responses to main points #4 and #5.

2) In Figure 3C legend, there is a typo in the sentence: {greater than or equal to}2 bins from a telomere. What does "bin" mean in the figure legend?

Bins refer to genomic segments used for Hi-C analysis, in this case of 32 kb in length. To clarify this, we now describe them as “genomic bins”.

3) Does the homologous pairing at HAS1pr-TDA1pr regulate nearby gene expression in saturated cultures? The authors could analyze the RNA-seq data to address this question.

Please see our responses to main points #1 and #7.

Reviewer #3:[…] - Subsection “A cluster of TF motifs is necessary and sufficient for HAS1pr-TDA1pr pairing” and Figure 1D: The authors claim to identify five clusters, however this choice seems arbitrary with additional depletions of substituted nucleotides (e.g regions upstream of E and between G-H).

We now selected a threshold of depletion of substitutions (log2 < -1.1) to highlight positions in Figure 1D based on the bimodal distribution of depletion scores (Figure 1—figure supplement F). Based on this analysis, we have further highlighted a sixth cluster that also matches the Rgt1 motif (Figure 1H). These highlighted motifs span all but one of the positions most depleted of mutations in the 3C library, and all clusters of two or more adjacent positions with extreme scores.

- Figure 1—figure supplement 2C: While the intensity of ratios is clearly less intense in the control than in the 3C library, it shows a similar pattern (compare panel C to panel B).

We note that while there are regions with weak enrichment or depletion in even the control 3C library, the positions with signal do not match those of the on-target 3C library.

- Subsection “Three transcription factors are required for pairing”, second paragraph is difficult to understand. How does the position of deleted regions influence the interactions? This should be improved together with the explanation of trans MAP-C and Figure 2A.

Apologies. We have sought to rewrite this section to explain these aspects more clearly.

- The identification of three transcriptional factors (TFs) required for HAS1pr-TDA1pr pairing suggests that transcriptional activity or RNA produced at the locus might play a role in regulating the interaction between the two regions. The authors should characterize the transcriptional status of the locus and its changes in the individual TFs knockouts and upon pairing.

Please see our responses to main points #1 and #7.

- Rgt1 domain analysis showed that the C-terminal region, largely structured, is required for pairing. However, there is quite a stretch from this observation to the suggestion that it is regulated by phosphorylation, as previously described (Polish et al., 2005). The authors should test the effect of mutating the regulatory S88 and S758 residues.

Please see our responses to main point #2.

[Editors' note: further revisions were requested prior to acceptance, as described below.][…] The manuscript has been improved but there is one remaining issue that needs to be addressed before acceptance. Reviewer 2 requests evidence that pairing has been successfully disrupted in order to validate interpretation of the results. Specifically, mutating Rgt1 binding sites results in a TDA1 upregulation but mutating Leu3 binding sites does not (Figure 5). It is important to determine by 3C-qPCR (for example) that pairing is lost in the mutants. The Log23C/genome data shown is not sufficient to make the claim. Reviewer 2 suggested that the authors compare HAS1pr-TDA1pr pairing with a non-pairing region or under exponential growth conditions. We believe that this experiment should not take too long to perform and would enable us to move forward with the manuscript, assuming all goes as planned.

As we discussed, we introduced highly targeted Leu3 and Rgt1 binding site mutations in a hybrid *S. cerevisiae* x *S. uvarum* context in an effort to rigorously test for transcriptional effects of pairing, rather than effects in *cis* or other effects in *trans*. We believe that these mutations do not completely disrupt pairing as reviewer #2 suggests, based on our *cis* and *trans* MAP-C experiments in which disrupting a single TF binding site or TF gene results in ~4-10-fold weaker pairing (Figure 1D, Figure 2B,C), compared to the ~30-fold increase in ectopic pairing (Figure 1C) conferred by the entire pairing sequence. Thus, as we have agreed upon, instead of performing the proposed experiment, we have edited the text (in the Results section and the Discussion section) to clarify our expectation that we have not fully disrupted pairing, but rather to ~10-25% of wild-type levels. We understand that reviewer #2 is concerned that this may have prevented us from finding transcriptional effects; however, this is just one of several potential reasons that we did not detect a phenotypic effect of disrupting *HAS1pr-TDA1pr* homolog pairing, as detailed in the Discussion section, and testing these possibilities is beyond the scope of this work.

Reviewer #2:The revised manuscript includes ChIP-seq and RNA-seq data to investigate the mechanism of HAS1pr-TDA1pr pairing. […] The authors should compare HAS1pr-TDA1pr pairing with a non-pairing region (or in exponential growth condition) as a control to show whether those mutants have entirely abolished the pairing or not.

See above.